# Learning Differential Pyramid Representation for Tone Mapping

**Qirui Yang** [1]    **Yinbo Li** [1]    **Yihao Liu** [2]    **Peng-Tao Jiang** [3]
**Fangpu Zhang** [1]    **Qihua Cheng**    **Huanjing Yue** [1]    **Jingyu Yang** [1*]

[1] School of Electrical and Information Engineering, Tianjin University, China
[2] Shanghai Artificial Intelligence Laboratory [3] vivo Mobile Communication Co., Ltd
{yangqirui, liyinbo, zhangfp, huanjing.yue, yjy}@tju.edu.cn,
liuyihao14@mails.ucas.ac.cn, pt.jiang@vivo.com

## Abstract

Existing tone mapping methods operate on downsampled inputs and rely on hand-crafted pyramids to recover high-frequency details. These designs typically fail to preserve fine textures and structural fidelity in complex HDR scenes. Furthermore, most methods lack an effective mechanism to jointly model global tone consistency and local contrast enhancement, leading to globally flat or locally inconsistent outputs such as halo artifacts. We present the Differential Pyramid Representation Network (DPRNet), an end-to-end framework for high-fidelity tone mapping. At its core is a learnable differential pyramid that generalizes traditional Laplacian and Difference-of-Gaussian pyramids through content-aware differencing operations across scales. This allows DPRNet to adaptively capture high-frequency variations under diverse luminance and contrast conditions. To enforce perceptual consistency, DPRNet incorporates global tone perception and local tone tuning modules operating on downsampled inputs, enabling efficient yet expressive tone adaptation. Finally, an iterative detail enhancement module progressively restores the full-resolution output in a coarse-to-fine manner, reinforcing structure and sharpness. Experiments show that DPRNet achieves state-of-the-art results, improving PSNR by **2.39 dB** on the 4K HDR+ dataset and **3.01 dB** on the 4K HDRI Haven dataset, while producing perceptually coherent, detail-preserving results. *We provide an anonymous online demo at DPRNet.*

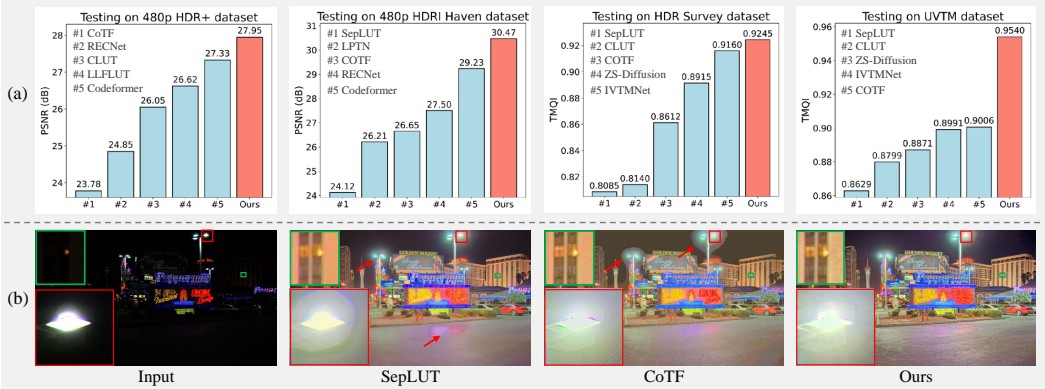

Figure 1: (a) Our DPRNet provides remarkable performance gains on the four datasets. (b) Existing tone mapping methods suffer from detail loss and halo artifacts.

*Corresponding author. This work is supported by the National Natural Science Foundation of China under Grants 62231018 and 62472308.

39th Conference on Neural Information Processing Systems (NeurIPS 2025).

# 1 Introduction

High dynamic range (HDR) imaging enables more faithful visual reproduction by capturing the wide luminance and color variations present in natural scenes. However, since most consumer displays support only a limited dynamic range, HDR content must be compressed via tone mapping to retain perceptual quality on standard devices. This compression task is inherently ill-posed: it requires reconciling global luminance consistency with the preservation of fine local structure and detail.

Many recent tone mapping methods adopt a low-resolution pipeline: they downsample the HDR input, perform tone compression at reduced scale, and then upsample the output to full resolution [1; 2; 3]. This strategy reduces computational overhead and allows the use of compact models. However, it introduces an intrinsic information bottleneck: once high-frequency details, such as fine edges, textures, and localized contrast, are lost during downsampling, they cannot be faithfully recovered by simple interpolation or shallow upsampling networks. This leads to visually flat results, especially in high-contrast or light regions, where subtle structural cues are critical.

To mitigate these losses, some methods [4; 5; 6] attempt to reintroduce high-frequency components using multi-scale decompositions, such as Laplacian or Gaussian pyramids [7; 8; 9]. These pyramids are handcrafted and fixed, applying the same filters across all scenes regardless of content, luminance dynamics, or semantic structure. Consequently, the extracted high-frequency representations are often misaligned with true image gradients or details. This mismatch can result in incom-

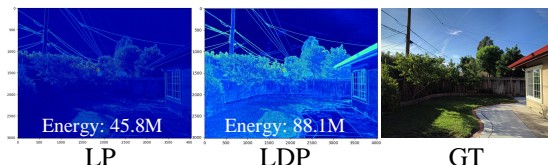

Figure 2: Example of high-frequency information extracted from the Laplacian pyramid (LP) and our LDP. We can observe that our LDP extracts high-frequency information more effectively than LP.

plete detail reconstruction and halo artifacts, particularly in HDR scenes where local contrast varies dramatically across spatial regions (see Fig. 2).

Beyond detail recovery, another core challenge lies in achieving tone consistency. Global tone mapping methods ensure uniform luminance but often suppress local contrast [10; 2], while local methods enhance fine detail at the risk of overexposure or inconsistency [11; 12]. Most learning-based tone mapping networks still treat global and local tone mapping as independent problems [13; 4], making it difficult to produce outputs that are both globally coherent and locally expressive.

In summary, current tone mapping frameworks face three fundamental limitations: **(1)** *High-frequency degradation* due to downsampling and weak reconstruction, (2) *Limited adaptability* of traditional multi-scale decompositions under diverse HDR conditions, and (3) *Disjoint modeling* of global tone control and local contrast adjustment, leading to incoherent perceptual results. **Our key insight** is that addressing these challenges requires a unified, learnable representation that bridges frequency-aware detail extraction with content-adaptive tone mapping.

To address these challenges, we propose DPRNet, an end-to-end *Differential Pyramid Representation Network* designed to tackle tone mapping comprehensively. At its core is a novel Learnable Differential Pyramid (LDP) that generalizes classical Difference-of-Gaussian pyramids with adaptive filtering and differencing, enabling robust high-frequency extraction across luminance scales. To enforce tonal consistency, we introduce a Global–Local Collaborative Tone Mapping mechanism consisting of: (1) a Global Tone Perception (GTP) module that processes downsampled inputs for scene-aware luminance adjustment, and (2) a Local Tone Tuning (LTT) module that refines patch-wise tone via learned 3D LUTs. Finally, an Iterative Detail Enhancement (IDE) module progressively integrates high-frequency components in a coarse-to-fine manner to reconstruct structurally faithful outputs. To facilitate robust training and benchmarking, we also introduce a new tone mapping dataset, HDRI Haven, that spans a wide range of real-world HDR scenes and lighting conditions.

In conclusion, the highlights of this work can be summarized into three points:

(1) We propose DPRNet, a unified tone mapping framework that integrates global tone perception, local tone adaptation, and high-frequency detail reconstruction.

(2) We design a Learnable Differential Pyramid for content-adaptive multi-scale detail extraction, and an Iterative Detail Enhancement module for progressive fidelity refinement.

(3) We introduce a new benchmark, HDRI Haven, enabling effective training and generalization.

(4) DPRNet achieves state-of-the-art performance, with **2.39 dB** PSNR improvement on the 4k HDR+ dataset and **3.01 dB** on the 4k HDRI Haven dataset, while producing perceptually coherent and detail-preserving results.

## 2 Related works

### 2.1 Learning-based Tone Mapping Methods

Recent advancements in tone mapping have leveraged deep learning techniques [6; 14] to address the nonlinear mapping from HDR to LDR images. Hou et al. [15] applied CNNs to tone mapping tasks, establishing a foundation for subsequent research. He et al. [10] developed a conditional sequential retouching network for effective image retouching. Similarly, Cao et al. [16], Rana et al. [17], and Panetta et al. [18] explored generative adversarial networks (GANs) for pixel-precise mapping. Despite these efforts, issues such as halo effects, noise, and local area processing remain challenging. To combat these issues, several recent works have introduced innovative solutions. For example, Hu et al. [19] combined tone mapping with denoising, incorporating a discrete cosine transform module for noise removal and improved image quality. Zhang et al. [20] manipulated tone in the HSV color space, which significantly reduced halos and preserved detail. Additionally, unsupervised and semi-supervised approaches have gained attention for their ability to handle large datasets without labeled data. Cao et al. [21] proposed an unsupervised HDR image and video tone mapping method using contrastive learning, which effectively captures the intrinsic properties of HDR images without explicit supervision. This method demonstrates strong generalization capabilities across different datasets. Zhu et al. [22] proposed a Diffusion-based [23] zero-shot tone mapping framework that utilizes shared structural knowledge to transfer pre-trained mapping models from the LDR domain to the HDR domain without paired training data. However, Diffusion-based methods are difficult to deploy effectively on mobile devices, limiting their range of applications.

For real-time applications, efficiency is a critical concern. Wang et al. [24] developed a fast global tone mapping algorithm for HDR compression, achieving real-time performance with minimal computational resources. Similarly, Zhang et al. [25] introduced a real-time semi-supervised deep tone mapping network that balances accuracy and speed, making it suitable for practical applications. In hardware implementations, latency and resource constraints pose additional challenges. Liang et al. [12] proposed a low-latency noise-aware tone mapping operator with a locally weighted guided filter, designed specifically for hardware implementation. This approach ensures high-quality tone mapping while maintaining low latency and computational efficiency. Despite the progress made by these learning-based methods, most of them are either global or local mappings, struggling to achieve satisfactory tone mapping results. To address this, our proposed Differential Pyramid Representation Network (DPRNet) combines global and local tone mapping while preserving fine details, offering a comprehensive solution to the challenges in HDR to LDR conversion.

### 2.2 3D LUT-based Enhancement Methods

In pursuit of computational efficiency and performance, emerging tone mapping methods employed 3D LUT [26; 27; 4] to render image brightness and color. Zeng et al. [2] developed an adaptive 3D LUT prediction network that merges multiple foundational 3D LUTs. Zhang et al. [3] presented a compressed representation of 3D LUTs, reducing parameters. Wang et al. [28] studied pixel-level fusion based on 3D LUTs, extending the method to a spatially aware variant. Yang et al. [29] proposed an effective ICELUT for extremely efficient edge inference. Yang et al. [30] introduced the AdaInt mechanism, which adaptively learns non-uniform sampling intervals in the 3D color space for more flexible sampling point allocation. Furthermore, Yang et al. [1] proposed SepLUT, addressing the shortcomings of 1D LUTs in color component interaction while alleviating the memory consumption issues of 3D LUTs. Despite these efforts, these methods necessitate an initial down-sampling step to reduce the network computation, and in the case of 4K images, even need 16 times down-sampling. As a result, this leads to a loss of image details and degradation of image quality.

## 3 Methods

### 3.1 Motivation

While tone mapping has advanced considerably, most existing methods prioritize computational efficiency by processing low-resolution representations of HDR images. These methods apply tone mapping to downsampled inputs, then upsample the outputs back to the original size. However, this

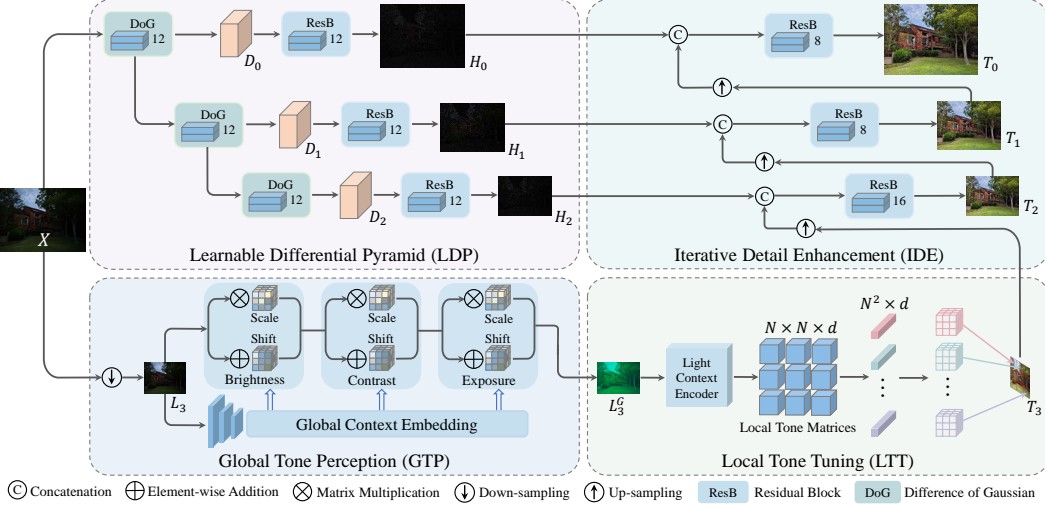

Figure 3: The overall pipeline of our DPRNet. It includes the proposed Learnable Differential Pyramid (LDP), Iterative Detail Enhancement (IDE), Global Tone Perception (GTP), and Local Tone Tuning (LTT). The LDP, in synergy with the IDE, is responsible for extracting and enhancing the high-frequency details. The GTP collaborates with LTT to achieve globally consistent yet locally fine-grained tone mapping.

approach often compromises high-frequency fidelity, leading to blurred textures, detail loss, and halo artifacts, particularly in high-contrast or brightly lit areas. To mitigate such losses, prior works [4; 31; 32] have incorporated multi-scale decomposition frameworks such as Laplacian pyramids [7] to recover fine details. Nevertheless, these pyramids rely on fixed, handcrafted filters that lack adaptability to the diverse luminance distributions and complex structures of HDR scenes. As a result, the high-frequency components extracted using such methods are often unstable or incomplete, ultimately resulting in over-smoothed reconstructions. *This highlights a critical need for a learnable, adaptive mechanism to robustly extract and preserve high-frequency features from HDR inputs.*

In parallel, effective tone mapping must also maintain tonal harmonization across the image. Global tone mapping [10; 2] enhances perceptual consistency, but often neglects localized details. In contrast, local methods [11; 12] refine contrast in specific regions, yet lack contextual awareness, which can lead to visual inconsistencies. The lack of integration between these two perspectives results in tone mapping systems that are either globally coherent but locally flat or locally sharp but globally inconsistent. *Thus, the central challenge is to design a unified tone mapping network that effectively captures high-frequency detail while jointly modeling both global and local tonal coherence.*

### 3.2 Framework Overview

To address these challenges, we propose DPRNet, an end-to-end network for HDR tone mapping that simultaneously achieves structural fidelity and tonal coherence. As illustrated in Fig. 3, DPRNet is composed of four main components: 1) the Learnable Differential Pyramid (LDP) for adaptive extraction of multi-scale high-frequency, 2) the Global Tone Perception (GTP) module to perform global luminance adaptation based on image-aware, 3) the Local Tone Tuning (LTT) module to refine regional contrast through patch-aware 3D LUTs, and 4) Iterative Detail Enhancement (IDE) module for progressive refinement of high-frequency content.

Given an HDR image $\mathbf{X} \in \mathbb{R}^{H \times W \times 3}$, our goal is to produce a perceptually faithful LDR output $\mathbf{T}_0$. The LDP first extracts multi-scale high-frequency components from $\mathbf{X}$. The GTP module then performs global tone mapping on a low-resolution version of the input ($\mathbf{L}_3$), while LTT applies localized adjustments to enhance regional contrast. Finally, IDE progressively integrates the high-frequency features into the tone-mapped image, refining detail from coarse to fine scales.

### 3.3 Learnable Differential Pyramid

High-frequency extraction is crucial for preserving detail in HDR images, which often contain rich textures and wider luminance range. To this end, we propose the LDP, a trainable, multi-scale decomposition module inspired by the principles of differential filtering and scale invariance [33; 34].

LDP exhibits two desirable properties: 1) it produces stable feature representations under scale and illumination changes, and 2) it captures edges, corners, and texture patterns effectively through learned difference operations.

The pipeline begins by convolving the input image $\mathbf{X}$ with a $3 \times 3$ kernel to generate an initial feature map $\mathbf{F}_0$. A lower-resolution map $\mathbf{F}_1$ is then produced via four convolutions and max pooling. To model the Difference of Gaussian (DoG) [35; 36], we compute the differences between the outputs of these four convolutions, resulting in a set of differential features $\mathbf{D}_0$. These differential features are then concatenated and passed through a residual block to produce the first high-frequency output $\mathbf{H}_0$. This process is repeated for three scales, yielding a differential pyramid:

$$\mathbf{X}_{\text{HF}} = \{\mathbf{H}_0, \mathbf{H}_1, \mathbf{H}_2\}, \tag{1}$$

with progressively reduced spatial resolutions from $H \times W$ to $\frac{H}{4} \times \frac{W}{4}$. These high-frequency features are used downstream to restore lost high-frequency detail in the final image.

### 3.4 Global-Local Collaborative Tone Mapping

An effective tone mapping system must address both global perceptual consistency and local contrast enhancement. To this end, we propose a collaborative tone mapping strategy that integrates a GTP module and an LTT module. The GTP collaborates with LTT to achieve image-wide tonal alignment yet locally coherent tone mapping. Their combination enables DPRNet to handle complex HDR scenes with rich structure and varying illumination.

**Global Tone Perception (GTP).** GTP performs global tone mapping on the downsampled input $\mathbf{L}_3 \in \mathbb{R}^{\frac{H}{8} \times \frac{W}{8} \times 3}$ to reduce computational load while capturing scene-wide illumination. It comprises two parts: a condition network $\mathcal{C}(\cdot)$ that extracts global context features, and a modulated convolutional core $\mathcal{F}(\cdot)$ that adapts global information (brightness, contrast, exposure) based on the input context. The condition network $\mathcal{C}(\cdot)$ consists of three convolutional layers, which extract global features, and apply global average pooling (GAP) to produce a condition vector $\mathbf{z}$:

$$\mathbf{z} = \text{GAP}(\mathcal{C}(\mathbf{L}_3)). \tag{2}$$

This vector is then used to generate per-layer modulation parameters via three fully connected layers:

$$\gamma^{(l)} = \mathbf{W}_\gamma^{(l)} \mathbf{z}, \quad \beta^{(l)} = \mathbf{W}_\beta^{(l)} \mathbf{z}, \quad l = 1, 2, 3, \tag{3}$$

where $\mathbf{W}_\gamma^{(l)}, \mathbf{W}_\beta^{(l)} \in \mathbb{R}^{d \times C_l}$ are learned weight matrices that generate modulation vectors for each layer $l$, and $C_l$ is the number of channels at layer $l$. Each layer's output is modulated as:

$$\mathbf{F}^{(l)} = \text{ReLU}\left(\mathbf{W}^{(l)} * \mathbf{F}^{(l-1)} \cdot \gamma^{(l)} + \beta^{(l)} + \mathbf{F}^{(l-1)}\right). \tag{4}$$

The final output is the globally tone mapped image $\mathbf{F}^{(3)} = \mathbf{L}_3^G$.

This conditional modulation allows GTP to adapt global information based on the scene context, leading to improved brightness and color consistency across varying global lighting conditions.

**Local Tone Tuning (LTT).** While GTP ensures global alignment, local details such as highlights, shadows, or complex textures still require region-specific enhancement. To address this, we introduce LTT, a region-aware tone adjustment module based on learned 3D LUTs. We first resize $\mathbf{L}_3^G$ and extract contextual features using a lightweight encoder $\mathcal{E}(\cdot)$:

$$\mathbf{F}_{\text{CE}} = \mathcal{E}(\mathbf{L}_3^G) \in \mathbb{R}^{B \times 6 \times 4 \times 4}, \tag{5}$$

where the lightweight encoder consists of three convolutional layers, LeakyReLU activation, instance normalization, and adaptive pooling. This contextual features $\mathbf{F}_{\text{CE}}$ is reshaped into $N^2$ local descriptors with dimensionality $d = 6$:

$$\left\{\mathbf{e}_i \in \mathbb{R}^6\right\}_{i=1}^{N^2}, \tag{6}$$

where $N = 4$ in our setting. Each local descriptor $\mathbf{e}_i$ is passed to a fully connected network (denoted FC) to predict weights over a bank of $R$ basis LUTs:

$$\mathbf{w}_i = \text{FC}(\mathbf{e}_i) \in \mathbb{R}^R. \tag{7}$$

Table 1: Quantitative results of TM methods on the HDR+ dataset. The "N.A." result is unavailable due to insufficient GPU memory. The "*" symbol indicates that the results are adopted from the original paper (some are absent ("/")) due to the unavailable source code. Metrics with ↑ and ↓ denote higher better and lower better. The best and second results are in red and blue, respectively.

| Method | #Params | HDR+ (480p) | | | | | HDR+ (original 4K) | | | | |
|---|---|---|---|---|---|---|---|---|---|---|---|
| | | PSNR↑ | SSIM↑ | TMQI↑ | LPIPS↓ | △E↓ | PSNR↑ | SSIM↑ | TMQI↑ | LPIPS↓ | △E↓ |
| UPE [37] | 999K | 23.33 | 0.852 | 0.856 | 0.150 | 7.68 | 21.54 | 0.723 | 0.821 | 0.361 | 9.88 |
| HDRNet [38] | 482K | 24.15 | 0.845 | 0.877 | 0.110 | 7.15 | 23.94 | 0.796 | 0.845 | 0.266 | 6.77 |
| CSRNet [10] | 37K | 23.72 | 0.864 | 0.884 | 0.104 | 6.67 | 22.54 | 0.766 | 0.850 | 0.284 | 7.55 |
| DeepLPF [39] | 1720K | 25.73 | 0.902 | 0.877 | 0.073 | 6.05 | N.A. | N.A. | N.A. | N.A. | N.A. |
| LUT [2] | 592K | 23.29 | 0.855 | 0.882 | 0.117 | 7.16 | 21.78 | 0.772 | 0.850 | 0.303 | 9.45 |
| LPTN [31] | 616K | 24.80 | 0.884 | 0.885 | 0.087 | 8.38 | 24.05 | 0.807 | 0.839 | 0.207 | 9.04 |
| CLUT [3] | 952K | 26.05 | 0.892 | 0.886 | 0.088 | 5.57 | 24.04 | 0.789 | 0.848 | 0.245 | 6.78 |
| sLUT [28]* | 4520K | 26.13 | 0.901 | / | 0.069 | 5.34 | 23.98 | 0.789 | / | 0.242 | 6.85 |
| SepLUT [1] | 120K | 22.71 | 0.833 | 0.879 | 0.093 | 8.62 | 21.87 | 0.731 | 0.842 | 0.220 | 9.52 |
| LLFLUT [4]* | 731K | 26.62 | 0.907 | / | 0.063 | 5.31 | 25.32 | 0.849 | / | 0.149 | 6.03 |
| CODEFormer [40] | 12232K | 27.33 | 0.941 | 0.882 | 0.028 | 5.79 | N.A. | N.A. | N.A. | N.A. | N.A. |
| CoTF [41] | 310K | 23.78 | 0.882 | 0.876 | 0.072 | 7.76 | 23.44 | 0.818 | 0.866 | 0.175 | 8.01 |
| RECNet [42] | 39K | 24.85 | 0.905 | 0.883 | 0.063 | 7.53 | 23.26 | 0.818 | 0.865 | 0.148 | 8.54 |
| Ours | 212K | 27.95 | 0.931 | 0.888 | 0.033 | 5.72 | 27.71 | 0.920 | 0.870 | 0.066 | 5.31 |

The final LUT for patch $i$ is constructed as a weighted sum:

$$\text{LUT}_i = \sum_{r=1}^{R} w_i^{(r)} \cdot \mathbf{B}^{(r)}, \tag{8}$$

where $\text{LUT}_i \in \mathbb{R}^{3 \times V^3}$ for patch-wise contrast enhancement, $\mathbf{B}^{(r)}$ denotes the $r$-th basis LUT (shared across patches), $V$ is the number of discretization points per channel (typically $V = 9$), and 3 corresponds to the RGB channels. To apply each LUT, we use a differentiable trilinear interpolation operator $\mathcal{I}(\cdot)$:

$$\mathbf{T}_i = \mathcal{I}(\mathbf{P}_i, \text{LUT}_i), \tag{9}$$

where $\mathbf{P}_i \in \mathbb{R}^{H_i \times W_i \times 3}$ is the $i$-th image patch and $\mathbf{T}_i$ is the tone-adjusted patch. To ensure spatial smoothness, bilinear blending is applied at patch boundaries during reassembly of the full image $\mathbf{T}_3$.

**Collaborative Mapping Strategy.** By applying GTP and LTT sequentially, DPRNet achieves globally consistent yet locally expressive tone mapping. GTP sets a stable tonal foundation, while LTT introduces fine-grained control over regional contrast and detail, resulting in images that are both perceptually coherent and rich in local structure.

### 3.5 Iterative Detail Enhancement

To restore fine details lost during downsampling or tonal transformations, DPRNet incorporates the IDE module. IDE integrates the previously extracted high-frequency features $\mathbf{X}_{\text{HF}}$ into the tone-mapped image $\mathbf{T}_3$ through a progressive, coarse-to-fine reconstruction process. At each scale $n$, the current output $\mathbf{T}_n$ is upsampled and concat with the corresponding high-frequency map $\mathbf{H}_{n-1}$:

$$\mathbf{F}_{n-1} = \text{Concat}(\text{Up}(\mathbf{T}_n), \mathbf{H}_{n-1}), \tag{10}$$

where $\text{Up}(\cdot)$ denotes the upsampling operation. A refinement network $\mathcal{R}^{(n-1)}(\cdot)$ then predicts a residual correction, added to the upsampled input:

$$\mathbf{T}_{n-1} = \text{Up}(\mathbf{T}_n) + \mathcal{R}^{(n-1)}(\mathbf{F}_{n-1}), \tag{11}$$

where $\mathcal{R}(\cdot)$ consists of residual blocks. This design enables adaptive integration of HF textures across multiple scales, yielding high-quality LDR outputs with enhanced sharpness and structural fidelity.

### 3.6 Loss Functions

To ensure accurate reconstruction of the image, we propose multi-loss functions that guides both pixel-level and scale-wise image enhancement. Our optimization jointly incorporates pixel-wise reconstruction loss $L_{\text{Re}}$, structural similarity loss $L_{\text{ssim}}$, high-frequency loss $L_{\text{HF}}$, and perceptual loss $L_{\text{p}}$. To summarize, the complete objective of our proposed model is combined as follows:

$$L_{\text{total}} = \alpha \cdot L_{\text{Re}} + \beta \cdot L_{\text{ssim}} + \gamma \cdot L_{\text{HF}} + \eta \cdot L_{\text{p}}, \tag{12}$$

where $\alpha$, $\beta$, $\gamma$, and $\eta$ are the corresponding weight coefficients. More details are in the appendix.

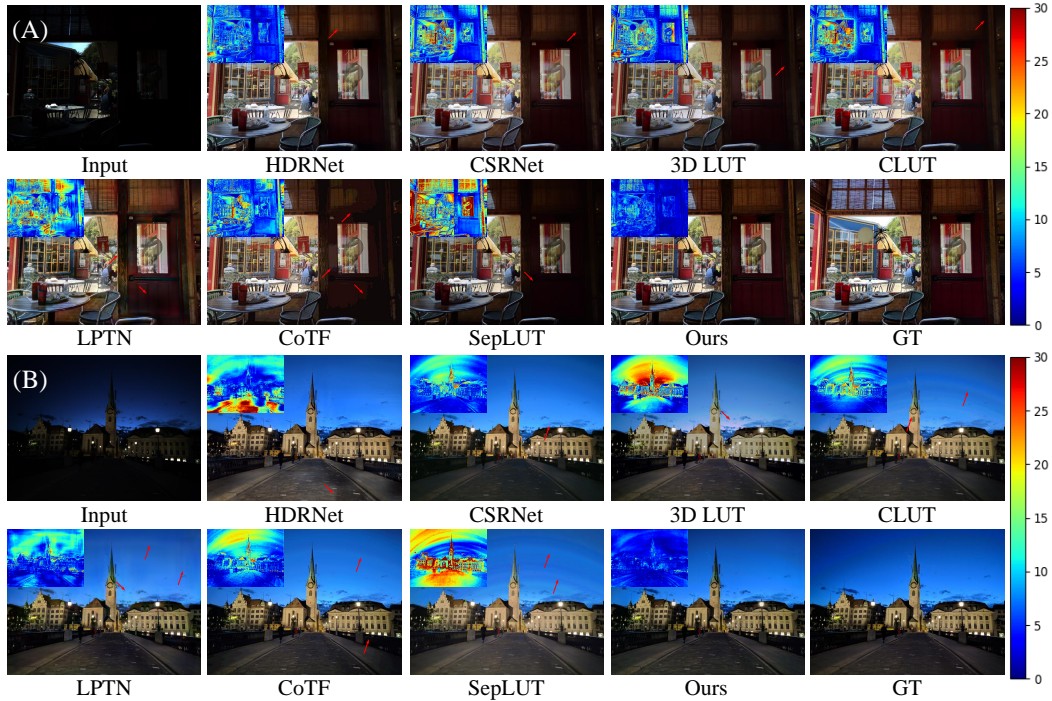

Figure 4: Visual comparisons between our DPRNet and the other methods on the 480p HDR+ dataset. The error maps in the upper left corner provide clearer insights into performance differences.

Table 2: Quantitative results on the HDRI Haven dataset. The "N.A." result is not available due to insufficient GPU memory.

| Method | HDRI Haven (480p) | | | | | HDRI Haven (original 4K) | | | | |
|---|---|---|---|---|---|---|---|---|---|---|
| | PSNR↑ | SSIM↑ | TMQI↑ | LPIPS↓ | △E↓ | PSNR↑ | SSIM↑ | TMQI↑ | LPIPS↓ | △E↓ |
| UPE [37] | 23.58 | 0.821 | 0.917 | 0.191 | 10.85 | 21.62 | 0.776 | 0.875 | 0.232 | 13.17 |
| HDRNet [38] | 25.33 | 0.912 | 0.941 | 0.113 | 7.03 | 23.61 | 0.899 | 0.890 | 0.143 | 8.19 |
| DeepLPF [39] | 24.86 | 0.939 | 0.948 | 0.077 | 7.64 | N.A. | N.A. | N.A. | N.A. | N.A. |
| CSRNet [10] | 25.78 | 0.872 | 0.928 | 0.153 | 6.09 | 24.42 | 0.863 | 0.875 | 0.174 | 6.83 |
| LUT [2] | 24.52 | 0.846 | 0.912 | 0.171 | 7.33 | 21.80 | 0.823 | 0.849 | 0.197 | 8.49 |
| CLUT [3] | 24.29 | 0.836 | 0.908 | 0.169 | 7.08 | 22.32 | 0.765 | 0.842 | 0.281 | 9.31 |
| LPTN [31] | 26.21 | 0.941 | 0.954 | 0.113 | 8.82 | 23.83 | 0.899 | 0.932 | 0.158 | 10.09 |
| SepLUT [1] | 24.12 | 0.854 | 0.915 | 0.165 | 8.03 | 22.79 | 0.838 | 0.859 | 0.180 | 9.11 |
| CODEFormer [40] | 29.23 | **0.967** | 0.956 | **0.021** | 4.81 | N.A. | N.A. | N.A. | N.A. | N.A. |
| CoTF [41] | 26.65 | 0.935 | 0.948 | 0.098 | 5.84 | 24.58 | 0.891 | 0.911 | 0.156 | 7.19 |
| RECNet [42] | 27.50 | 0.941 | 0.955 | 0.054 | 6.66 | 25.29 | 0.898 | 0.907 | 0.126 | 7.06 |
| Ours | **30.47** | 0.966 | **0.959** | 0.024 | **4.62** | **28.30** | **0.946** | **0.938** | **0.034** | **5.43** |

## 4 Experiments

### 4.1 Experimental Settings

**Datasets.** We evaluate our method on four diverse datasets: HDR+ Burst Photography [43], HDRI Haven, HDR Survey [44], and the UVTM video dataset [21]. The HDR+ dataset, commonly used for HDR reconstruction and tone mapping research, includes 675 image sets for training and 248 for testing at both 480p and 4K resolutions, following the preprocessing method of Zeng et al. [2]. For HDRI Haven, we collected 570 HDR image pairs from the website HDRI Haven and created corresponding ground truths using several software and toning tools combined with selection algorithms. These images cover a wide range of scenes, such as indoor and outdoor, including natural light/artificial light, multi-contrast, sunrise/sunset, urban/nature, daytime/nighttime, and so on. We select 456 image sets for training and 114 for testing. The HDR Survey dataset consists of 105 real HDR images, with no ground truth, and is one of the benchmarks for HDR tone mapping evaluations [21; 22]. Lastly, the UVTM video dataset, also with no ground truth, includes 20 real captured HDR videos. Note that the HDR Survey and UVTM video datasets are for real-world testing only.

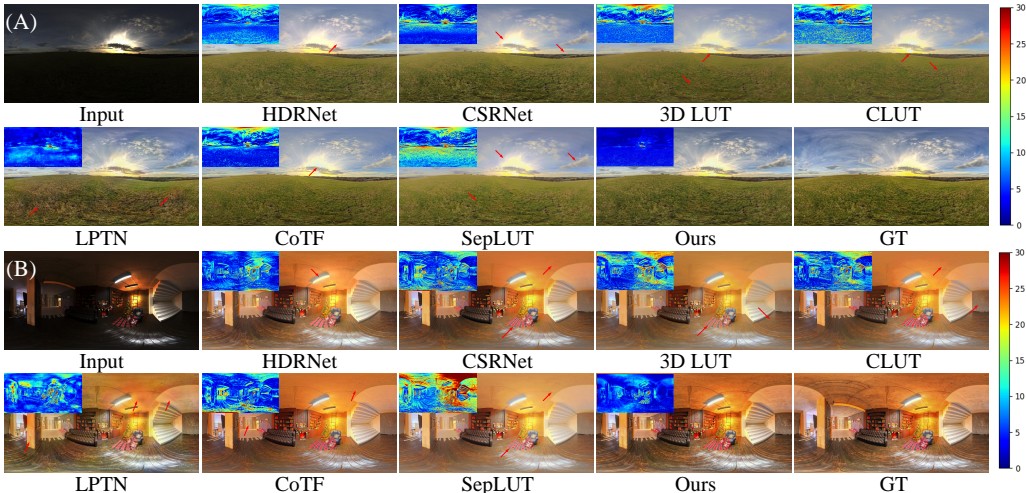

Figure 5: Visual comparisons between our DPRNet and the state-of-the-art methods on the 480p HDRI Haven dataset (Zoom-in for best view).

**Implementation Details.** We evaluate model performance in five dimensions: pixel accuracy (PSNR), structural integrity (SSIM), mapping quality (TMQI [45]), perceptual quality (LPIPS [46]), and color fidelity ($\Delta E$ [47]). We implement the model using PyTorch on an RTX 3090 GPU and optimize using the AdamW optimizer [48], with $\beta_1 = 0.9$, $\beta_2 = 0.99$, and an initial learning rate of $1 \times 10^{-4}$.

## 4.2 Comparison with State-of-the-Arts

**Quantitative Comparison.** We compare DPRNet with several state-of-the-art tone mapping methods that use HDR+ dataset as benchmarks, including LLFLUT [4], SepLUT [1], CLUT [17], sLUT [28], LUT [2], and HDRNet [38]. We further add comparison with image enhancement and exposure correction methods, including CoTF [41], RECNet [42], ZS-Diffusion [22], IVTMNet [21], CODEFormer [40], UPE [37], CSRNet [10], DeepLPF [39], LPTN [31]. To ensure a fair comparison, we follow the results in the latest published papers [4] and use publicly available source code and recommended configurations for training other methods.

Our method shows a significant improvement over the most recent works, LLFLUT (NeurIPS 2023 [4]) and RECNet (AAAI 2024 [42]). Tab. 1 reports the results for the HDR+ dataset at both 480p and 4K resolutions. DPRNet demonstrates robust performance at higher resolutions, with a performance boost in PSNR of **2.39 dB** compared to LLFLUT at original resolution (4K). On the HDRI Haven dataset, as shown in Tab. 2, DPRNet achieves over **3 dB** improvement at 4K resolution, reflecting the fine balance between global and local tone mapping in our framework. Compared to previous methods, our DPRNet demonstrates superiority with acceptable parameters (**212K**), further demonstrating the superiority of our approach.

**Qualitative Results.** Visual comparisons between DPRNet and state-of-the-art methods are shown in Fig. 4, Fig. 5, and Fig. 7. To better highlight the performance differences, we provide error maps where red areas represent larger discrepancies, and blue areas indicate closer alignment to the target image. These visual comparisons consistently show that DPRNet produces superior results in terms of local detail preservation, color fidelity, and structural integrity across HDR+ and HDRI Haven datasets. For example, in Fig. 4, DPRNet excels in local detail processing while maintaining color accuracy, demonstrating a better balance between global and local regions. In contrast, previous methods either result in color distortion (e.g., 3D LUT and SepLUT in Fig. 4(A), LPTN and CSRNet in Fig. 5(A)) or fail to map local area (e.g., HDRNet, CSRNet, CoTF, SepLUT in Fig. 4(B), CLUT, SepLUT in Fig. 5(B). For real-world scenes, as shown in Fig. 7, our method faithfully reconstructs fine high-frequency textures while ensuring precise color reconstruction and vivid color saturation.

## 4.3 Ablation Studies

**Effectiveness of Specific Modules.** To evaluate the impact of each key component in our proposed DPRNet, we conduct ablation experiments by systematically removing or modifying specific modules: LTT, GTP, LDP, and IDE. The results, summarized in Tab. 3, highlight the contribution of each

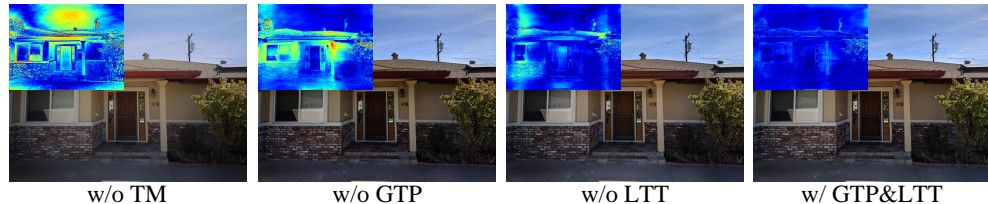

| w/o TM | w/o GTP | w/o LTT | w/ GTP&LTT |

Figure 6: Visual results of ablation study on GTP and LTT components.

module to the overall performance. From the table, we observe that removing the LTT module (Variant #1) results the PSNR drops to 26.36 dB from 27.95 dB (Variant #5), and the SSIM decreases from 0.931 to 0.921, showing that the local adjustment of tonal details is crucial for improving both the structural integrity and visual quality of the tone-mapped image. Removing GTP (Variant #2) also leads to a drop in performance. This indicates that global luminance adaptation plays a key role in ensuring the image maintains overall tonal consistency across varying lighting conditions, even though the local refinement (LTT) still contributes significantly.

As shown in Fig. 6, the collaboration of GTP with LTT yields promising visual quality in both global and local enhancement in this challenging case. When the LDP is removed (Variant #3), the performance shows a more substantial drop in PSNR (25.90 dB),

Table 3: Ablation studies of key components.

| Variants | LTT | GTP | LDP | IDE | PSNR↑ | SSIM↑ | TMQI↑ | LPIPS↓ | △E↓ |
|---|---|---|---|---|---|---|---|---|---|
| #1 | ✗ | ✓ | ✓ | ✓ | 26.36 | 0.921 | 0.887 | 0.042 | 6.50 |
| #2 | ✓ | ✗ | ✓ | ✓ | 26.17 | 0.913 | 0.885 | 0.043 | 6.73 |
| #3 | ✓ | ✓ | ✗ | ✓ | 25.90 | 0.912 | **0.889** | 0.076 | 8.61 |
| #4 | ✓ | ✓ | ✓ | ✗ | 25.35 | 0.872 | 0.877 | 0.071 | 6.85 |
| #5 | ✓ | ✓ | ✓ | ✓ | **27.95** | **0.931** | 0.888 | **0.033** | **5.72** |

which reflects a loss in detail fidelity. The PSNR degradation demonstrates the importance of the differential pyramid in capturing high-frequency details essential for fine texture preservation. Removing the IDE module (Variant #4) also leads to a significant decline in performance. Results demonstrate the importance of progressively refining high-frequency details for structural integrity. When all modules are combined (Variant #5), DPRNet achieves the highest performance. These results confirm the complementary roles of each component in enhancing the perceptual quality, structural detail, and tonal consistency of the final output.

In summary, the ablation study highlights the effectiveness of each component in DPRNet. Removing any single module results in a noticeable decrease in performance. The full model, integrating all modules, achieves the best overall performance, underscoring the importance of the collaborative interaction between LTT, GTP, LDP, and IDE in our framework.

**Ablation Study on Loss Functions.** We conduct an ablation study on the loss function to evaluate the impact of the loss function on training DPRNet. The results, summarized in Tab. 4, provide insights into the contribution of each loss to the overall performance. The ablation study reveals that each component of the loss function contributes to improving different

Table 4: Ablation study on the loss function.

| Variants | $L_{\text{Re}}$ | $L_{\text{ssim}}$ | $L_{\text{HF}}$ | $L_{\text{p}}$ | PSNR↑ | SSIM↑ |
|---|---|---|---|---|---|---|
| #1 | ✓ | ✗ | ✓ | ✓ | 27.85 | 0.929 |
| #2 | ✓ | ✓ | ✗ | ✓ | 27.74 | 0.927 |
| #3 | ✓ | ✓ | ✓ | ✗ | 27.76 | 0.925 |
| #4 | ✓ | ✓ | ✓ | ✓ | 27.95 | 0.931 |

aspects of the tone-mapped image. The $L_{\text{Re}}$ loss ensures pixel-wise accuracy, the $L_{\text{ssim}}$ loss enhances structural similarity, the $L_{\text{HF}}$ loss preserves fine details, and the $L_{\text{p}}$ loss improves visual quality. The best performance is achieved when all components are combined, underscoring the importance of each loss in achieving high-quality tone mapping.

**HDR+ or HDRI Haven Datasets better?** To evaluate the generalization capability of our HDRI Haven dataset, we conducted an ablation study comparing its performance against the HDR+ dataset. Specifically, we trained our model on both datasets and tested the performance on two real-world tone mapping datasets: HDR Survey

Table 5: Ablation study on the dataset.

| Method | Training Dataset | Testing Dataset | |
|---|---|---|---|
| | | HDR Survey [44] | UVTM [21] |
| DPRNet | HDR+ [43] | 0.8157 | 0.8476 |
| | HDRI Haven | **0.9245** | **0.9540** |

[44] and UVTM [21]. We used TMQI as the evaluation metric to assess the perceptual quality and consistency of the tone mapping results. As shown in Tab. 5, when trained on the HDR+ dataset, DPRNet achieves TMQI scores of 0.8157 on the HDR Survey dataset and 0.8476 on the UVTM dataset. However, when trained on the HDRI Haven dataset, DPRNet significantly outperforms this, achieving TMQI scores of **0.9245** and **0.9540**, respectively. These results demonstrate that

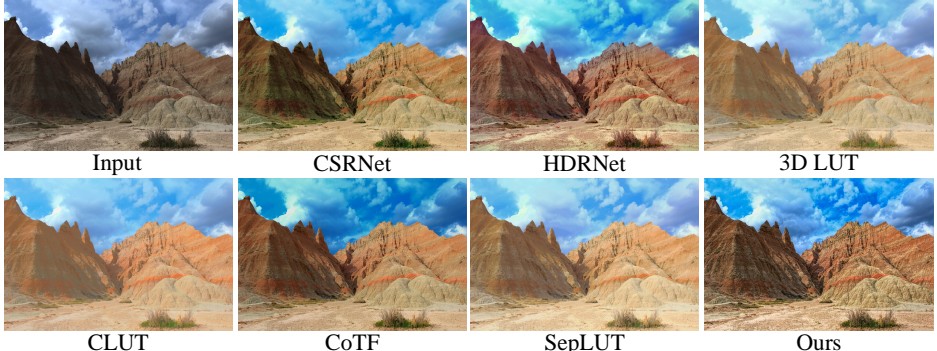

Figure 7: Visual comparisons between our DPRNet and the other methods.

HDRI Haven provides better generalization and is more robust for real-world HDR tone mapping, confirming its superior versatility and performance.

## 4.4 Validation of Generalization

We evaluate the generalization performance of the proposed model on the HDR Survey dataset [44] and UVTM video dataset [21] using the no-reference metric TMQI as the

Table 6: Validating generalization on third-party datasets, including HDR Survey and UVTM video datasets. IVTMNet and ZS-Diffusion report results from the original paper.

| Datasets | Metrics | 3D LUT | CLUT | SepLUT | IVTMNet [21] | CoTF | ZS-Diffusion [22] | Ours |
|---|---|---|---|---|---|---|---|---|
| HDR Survey | TMQI | 0.8165 | 0.8140 | 0.8085 | 0.9160 | 0.8612 | 0.8915 | **0.9245** |
| UVTM | TMQI | 0.8787 | 0.8799 | 0.8629 | 0.8991 | 0.9006 | 0.8871 | **0.9540** |

evaluation metric. We use the models trained on the 4K HDRI Haven dataset and report the results in Tab. 6. Our proposed model achieves the highest TMQI scores and significantly outperforms the other methods. Compared with the ZS-Diffusion [22], DPRNet achieves notable improvements of **0.033**, demonstrating its superiority in handling HDR content across diverse scenarios and datasets. Visual results in Fig. 7 show the superiority of the generalization performance of our model.

## 4.5 User Study

To evaluate the overall performance of the photorealism and visual quality, we perform a user study based on human perception. We conducted a subjective evaluation of 25 users with varying experiences with HDR processing. The participants are asked to rank four tone-mapped images (HDRNet [38], CLUT [3], GT, and Ours) according to the aesthetic visual quality. 50 images are randomly selected from the testing set and are shown to each participant. Four tone mapping results are displayed on the screen in a random order. Users are asked to pay attention to the color, details, artifacts, and whether the local color is harmonious. As shown in Fig. 8, our results achieve a better visual ranking compared to HDRNet and CLUT. Also, we got 413 images ranked first and

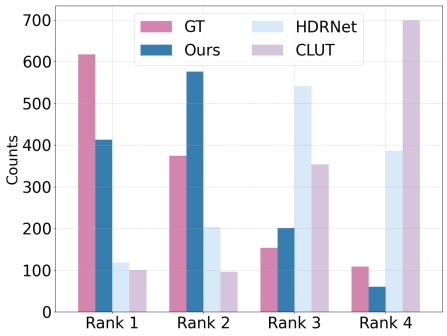

Figure 8: Ranking results of the user study. Rank 1 means the best visual quality.

576 images ranked second, accounting for **39.56%** of the total, achieving results comparable to the ground truth (accounting for 39.64%).

## 5 Conclusion

This paper proposes a learnable Differential Pyramid Representation Network (DPRNet), a comprehensive framework capable of recovering details and manipulating global and local tone mappings. We propose the LDP module to capture multi-scale high-frequency components from input HDR images. To achieve global consistency and local contrast harmonization, we embed a GTP and an LTT module within the DPRNet. Furthermore, we propose an IDE to gradually improve the image details by progressively refining the extracted high-frequency components. Extensive experiments demonstrate that our method significantly outperforms state-of-the-art methods, improving PSNR by 2.39 dB in the 4k HDR+ and 3.01 dB in the 4k HDRI Haven dataset, respectively, compared with the second-best method. In addition, our method has the best generalization ability in real-world video and image tone mapping.

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

# Appendix

## Contents

In this appendix, we present related work and provide additional ablation studies, qualitative results, and a discussion of limitations.

## A   Loss Functions

The proposed framework obtains faithful global and local enhancement by optimizing the reconstruction loss, perceptual loss, and high-frequency loss.

**Reconstruction Loss.** To maintain the accuracy of the reconstructed image, we directly adopt pixel-wise $L_{\mathrm{Re}}$ and $L_{\mathrm{ssim}}$ loss on the prediction $\mathbf{T}$ and the ground truth $\mathbf{Y}$:

$$L_{\mathrm{Re}} = \sum_{i=1}^{n} \left\| \mathbf{T}_i - \mathbf{Y}_i^{\mathrm{LF}} \right\|_1 + \left\| \mathbf{T} - \mathbf{Y} \right\|_1, \tag{13}$$

$$L_{\mathrm{ssim}} = 1 - \mathrm{MS\text{-}SSIM}(\mathbf{T}, \mathbf{Y}), \tag{14}$$

where $\mathbf{T}_i$ denotes the output of each layer of the network and $\mathbf{Y}_i$ denotes the Gaussian pyramid of the ground truth.

**High-Frequency Loss.** To prompt the learnable differential pyramid module to extract effective high-frequency information, we introduce a high-frequency loss function. By calculating the $L_1$ loss between the output high-frequency component and the high-frequency of ground truth:

$$L_{\mathrm{HF}} = \sum_{i=0}^{n-1} \left\| \mathbf{H}_i - \mathbf{Y}_i^{\mathrm{HF}} \right\|_1, \tag{15}$$

where $\mathbf{Y}_i^{\mathrm{HF}}$ denotes the high-frequency component of the GT obtained through the Laplacian pyramid.

**Perceptual Loss.** To make the learned DPRNet more stable and robust, we employ a perceptual loss function that assesses a solution concerning perceptually relevant characteristics (e.g., the structural contents and detailed textures):

$$L_{\mathrm{p}} = |\varphi^i(\mathbf{T}) - \varphi^i(\mathbf{Y})\|_2^2, \tag{16}$$

where $\varphi$ represents the 5th convolution (before maxpooling) layer within VGG19 network [49].

**Output Loss.** To summarize, the complete objective of our proposed model is combined as follows:

$$L_{\mathrm{total}} = \alpha \cdot L_{\mathrm{Re}} + \beta \cdot L_{\mathrm{ssim}} + \gamma \cdot L_{\mathrm{HF}} + \eta \cdot L_{\mathrm{p}}, \tag{17}$$

where $\alpha$, $\beta$, $\gamma$, and $\eta$ are the corresponding weight coefficients.

## B   More Ablation Studies

### B.1   Selection of the Number of Levels.

We validate the influence of the number of pyramid levels $n$. As shown in Tab. 7, the model achieves the best performance on all tested resolutions when $n = 4$. When a larger number of levels ($n \geq 5$) results in a significant decline in performance. This is because when $n$ is larger and the number of downsamples is greater, the model fails to reconstruct the high frequencies efficiently, resulting in performance degradation.

### B.2   Ablation Study on Size of the 3D LUTs.

To investigate the impact of 3D LUT (Look-Up Table) size in the Local Tone Tuning (LTT) module, we conduct an ablation study by varying both the number of dimensions in the LUT ($m$) and the number of discrete levels ($S_t$). Specifically, we examine two values for $m$ (6 and 8) and two values for $S_t$ (9 and 17), which define the granularity and the size of the LUT used for local tonal adjustments. The results are summarized in Tab. 8.

In summary, the ablation study indicates that $m = 6$ and $S_t = 9$ strikes the optimal balance between performance and computational efficiency, offering the best combination of image quality and structural similarity. Increasing $S_t$ or $m$ results in diminishing returns in performance, with a significant increase in computational cost. Therefore, the choice of LUT size should be carefully considered to maintain efficiency while maximizing performance.

Table 7: Ablation study on the number of levels.

| Metrics | Number of Levels | | |
|---|---|---|---|
| | n=3 | **n=4** | n=5 |
| PSNR | 27.11 | 27.95 | 27.02 |
| SSIM | 0.9179 | 0.9313 | 0.9109 |
| TMQI | 0.8898 | 0.8883 | 0.8857 |
| LPIPS | 0.0387 | 0.0335 | 0.0428 |
| $\triangle$E | 6.493 | 5.721 | 6.497 |

Table 8: Ablation study on size of the Look-Up Table (3D LUT).

| $m$ | $S_t$ | PSNR | SSIM | #Params |
|---|---|---|---|---|
| **6** | **9** | 27.95 | 0.931 | 212K |
| 8 | 9 | 27.92 | 0.926 | 214K |
| 6 | 17 | 27.98 | 0.927 | 815K |
| 8 | 17 | 27.71 | 0.928 | 816K |

## B.3 Selection of the Number of Grids.

To evaluate the impact of the number of grids (denoted as **N**) used in the Local Tone Tuning (LTT) module, we conduct an ablation study by varying the grid size. Specifically, we examine four different grid sizes: N=2, N=4, N=6, and N=8. The results are summarized in Tab. 9, which reports performance across several metrics, including PSNR, SSIM, TMQI, LPIPS, color difference ($\Delta E$), and the number of parameters in the model. As observed in Tab. 9, the N=4 strikes the optimal balance between model performance and computational efficiency. It achieves the best results across PSNR, SSIM, TMQI, LPIPS, and color fidelity, while maintaining a manageable model size. Larger grid sizes (N=6 and N=8) yield slightly worse performance, and smaller grids (N=2) do not capture enough local detail for high-quality tone mapping.

Table 9: Ablation study on the number of grids in the LTT.

| Metrics | Grid Size | | | |
|---|---|---|---|---|
| | N=2 | N=4 | N=6 | N=8 |
| PSNR | 25.77 | **27.95** | 27.03 | 27.41 |
| SSIM | 0.9146 | **0.9313** | 0.9203 | 0.9208 |
| TMQI | 0.8857 | **0.8883** | 0.8826 | 0.8855 |
| LPIPS | 0.0405 | **0.0335** | 0.0382 | 0.0373 |
| $\triangle$E | 6.866 | **5.721** | 6.190 | 5.967 |
| #Params | **135K** | 212K | 346K | 530K |

## B.4 Quantitative Comparisons on High-Frequency Components

To further validate the effectiveness of Learnable Differential Pyramid (LDP) in high-frequency reconstruction, we conducted a quantitative comparison with other methods using high-frequency components extracted via Laplacian decomposition. Specifically, we measured the PSNR and SSIM of the extracted high-frequency (HF) maps, and additionally computed the Edge Preservation Index (EPI) using the Canny edge detector to assess the fidelity of fine structural details.

As shown in Table 10, our method achieves the highest PSNR-HF (25.92) and SSIM-HF (0.8637) among all methods, indicating superior reconstruction accuracy and structural consistency in high-frequency regions. While CODEFormer achieves the highest EPI (0.8333), our method closely follows with an EPI of 0.8201, demonstrating strong edge retention capabilities. In contrast, traditional LUT-based methods (e.g., 3DLUT, SepLUT) perform significantly worse across all three metrics, highlighting the advantage of LDP in recovering fine-grained textures. These results confirm that our LDP effectively preserves high-frequency details and enhances fine structure reconstruction, outperforming both conventional and existing learning-based tone mapping approaches.

## B.5 More Generalization Comparisons

To verify the difference in generalization between the two datasets, we use the HDR+ data and the HDRI Haven dataset to train separately and test on the HDR Survey dataset [44]. We use the no-reference metric TMQI as the evaluation metric. Tab. 11 shows that training on the HDRI Haven dataset is more suitable for the tone mapping task than HDR+, providing a new benchmark for subsequent tone mapping tasks. Meanwhile, the results in Tab. 11 show that our proposed DPRNet has stronger model generalization performance.

Table 10: Quantitative Comparisons on High-Frequency Components.

| Methods | PSNR-HF | SSIM-HF | EPI |
|---|---|---|---|
| 3DLUT | 22.50 | 0.7436 | 0.6600 |
| CLUT | 22.79 | 0.7529 | 0.6745 |
| CSRNet | 22.71 | 0.7556 | 0.6704 |
| HDRNet | 24.02 | 0.8124 | 0.7569 |
| SepLUT | 22.22 | 0.7216 | 0.6491 |
| LPTN | 23.26 | 0.8079 | 0.7332 |
| COTF | 23.14 | 0.742 | 0.7241 |
| CODEFormer | 25.29 | 0.8592 | **0.8333** |
| RECNet | 24.10 | 0.8248 | 0.7626 |
| Ours | **25.92** | **0.8637** | 0.8201 |

Table 11: Differences in generalization between the HDR+ Dataset and the HDRI Haven dataset.

| Train Dataset | Test Dataset | TMQI | | | | | |
|---|---|---|---|---|---|---|---|
| | | CSRNet | 3D LUT | CLUT | LPTN | CoTF | Ours |
| HDR+ dataset | HDR Survey | 0.7754 | 0.7847 | 0.7638 | 0.7892 | 0.7959 | 0.8157 |
| HDRI Haven dataset | HDR Survey | 0.8439 | 0.8165 | 0.8140 | 0.8964 | 0.8612 | 0.9245 |

## C DPRNet for HDR Video Tone Mapping

In this section, we further demonstrate the effectiveness and model generalization performance of the proposed DPRNet for video tone mapping. We use the HDRI Haven image dataset for training and testing on the UVTM video dataset [50]. For display purposes, we have created an anonymous web page showing our results on the video dataset (DPRNet). As can be observed from the four sets of videos shown, our method generates videos with suitable contrast between light and dark and does not suffer from any unnatural textures. As shown in Fig. 10, our method exhibits perfect tone mapping results in different sophisticated scenes, proving the effectiveness of our proposed DPRNet and the strong model generalization ability.

## D More Qualitative Results

We provide additional qualitative comparisons on the HDR Survey, UVTM video, HDRI Haven, and HDR+ datasets. Fig. 9 to Fig. 14 show 17 sets of qualitative comparisons. It is observed that DPRNet successfully produces outputs with finer details and sharper edges. Furthermore, with the proposed joint global and local tone mapping, DPRNet further improves the color saturation of the output results. It can be observed that our proposed method achieves the best TMQI scores. For TMQI, our method outperforms the second-best method [31] by 0.053 on the UVTM dataset. It demonstrates that our method is good at revealing details and keeping temporal consistency. Fig. 9 shows four sets of qualitative comparisons on the HDR Survey dataset. From the examples, we see that DPRNet can restore the fine details, leading to plausible results. Notably, we train on the HDRI Haven dataset and test on the HDR Survey dataset and the UVTM video dataset. This proves that our DPRNet has not only strong robustness but also strong generalization performance.

## E Limitations and Discussion

Our method does have limitations. For the MIT FiveK dataset, our method is unable to obtain SOTA results, and we have performed some analyses on this. Firstly, FiveK was shot by a DSLR camera, which was not processed by an advanced ISP pipeline and lacked raw denoising and YUV denoising, resulting in the dataset containing serious noise. These noises have a great influence on our high-frequency extraction. Second, some reference images in the FiveK dataset suffer from overexposure or oversaturation, which poses a challenge to the enhancement method, as mentioned by Zhang et al [4]. Thirdly, there are inconsistencies in the reference images adjusted by the same professional photographer, leading to differences between the training and test sets. As a future work, we plan to build a unified model to tackle more visual enhancement tasks by integrating denoising and tone mapping.

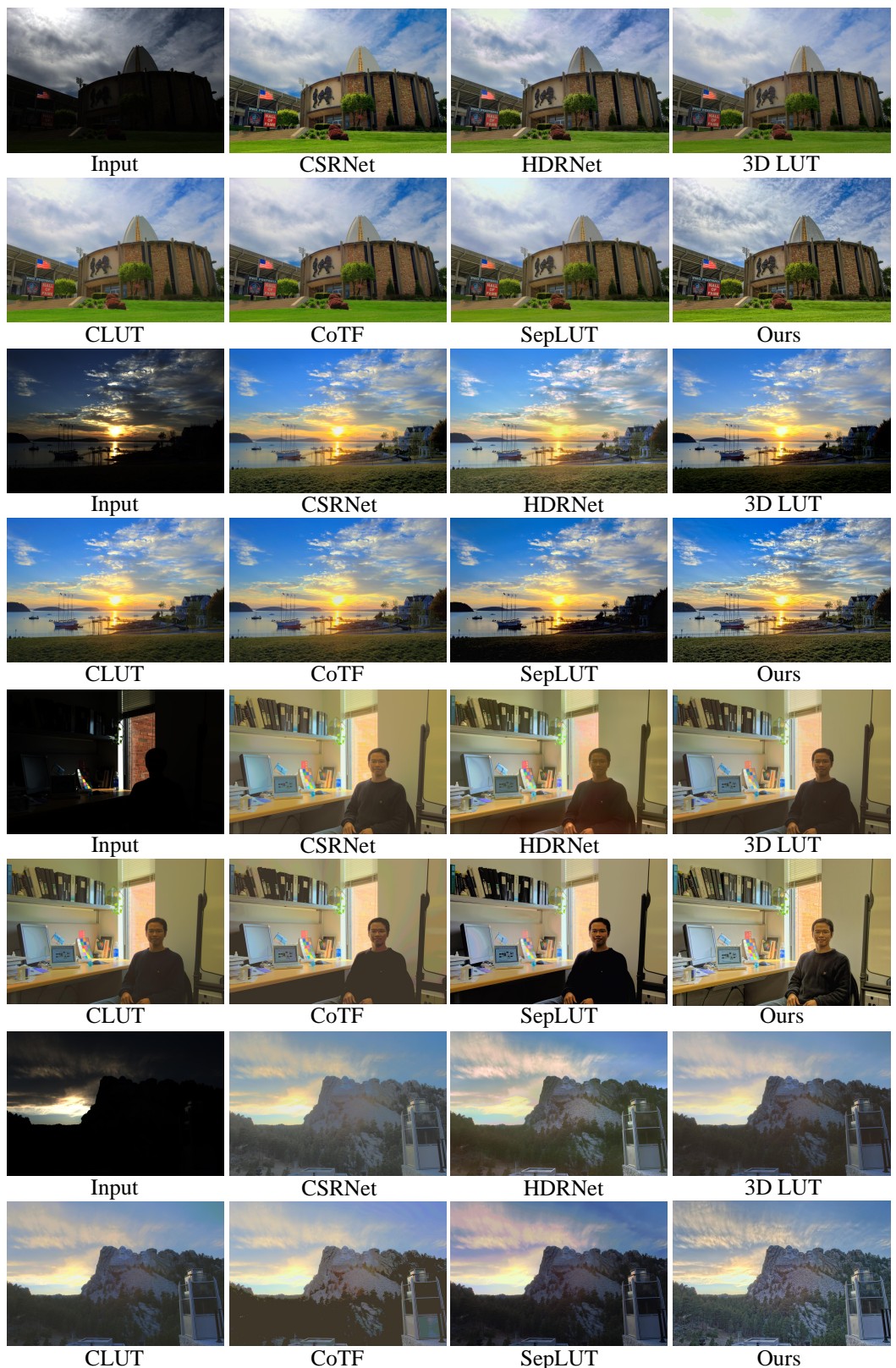

Figure 9: Visual comparisons between our DPRNet and the state-of-the-art methods, training on the HDR Haven dataset, and testing on the HDR Survey dataset (real-world dataset).

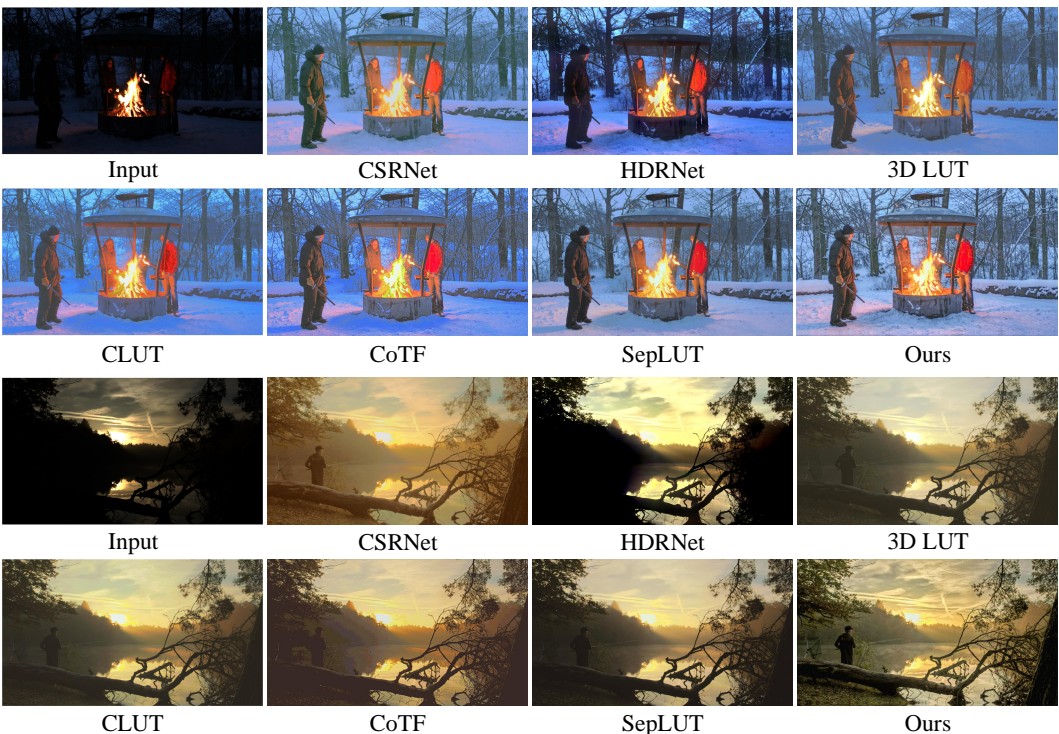

Figure 10: Visual comparisons between our DPRNet and the state-of-the-art methods, training on the HDR Haven dataset, and testing on the UVTM video dataset [21] (real-world dataset).

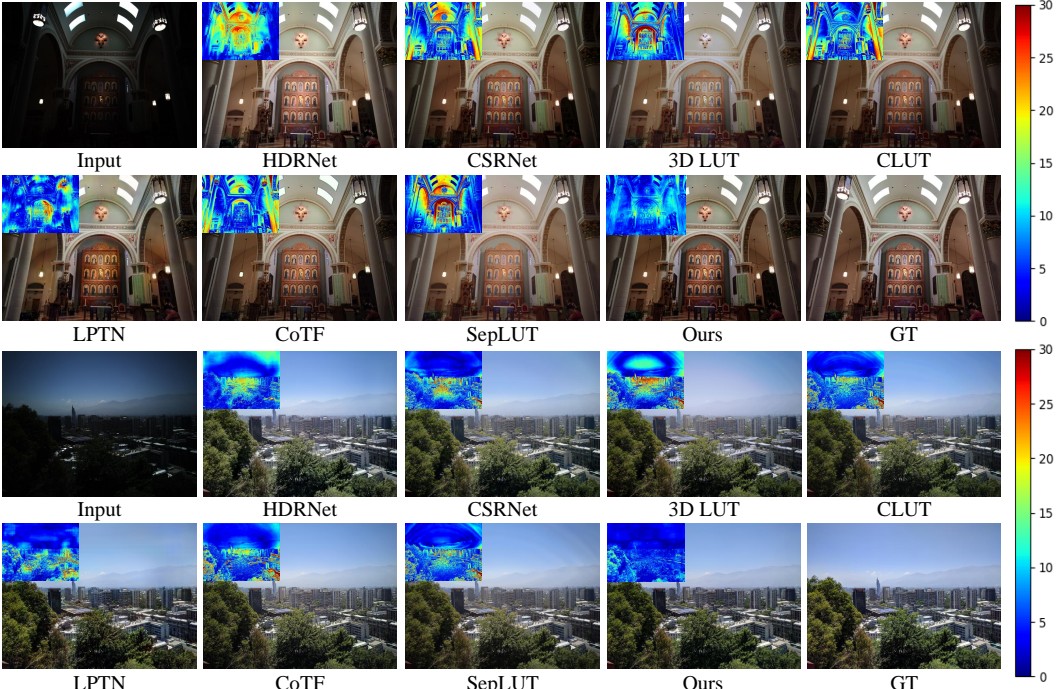

Figure 11: Visual comparisons between our DPRNet and the state-of-the-art methods on the HDR+ dataset (480p resolution).

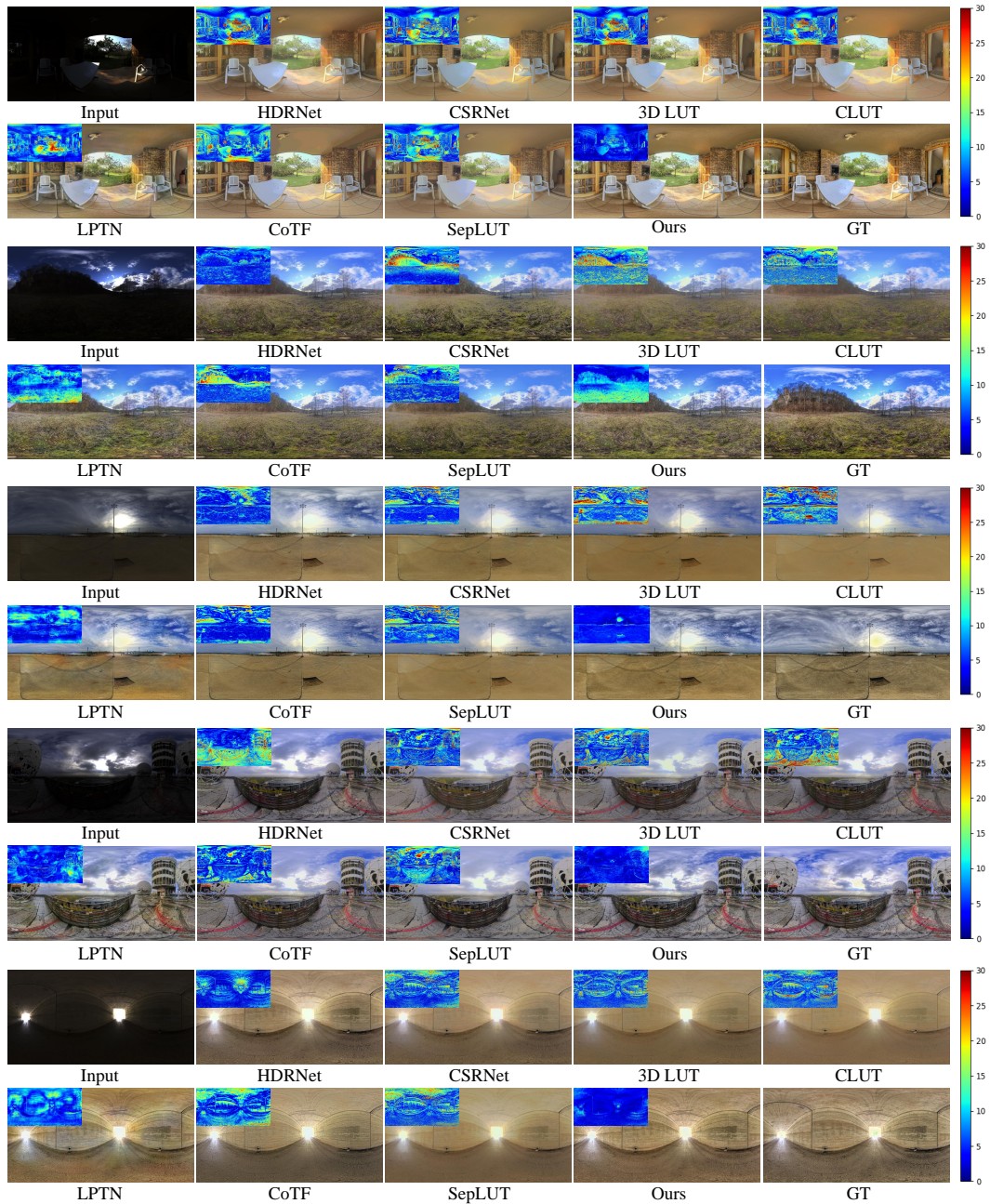

Figure 12: Visual comparisons between our DPRNet and the state-of-the-art methods on the HDRI Haven dataset (480p resolution).

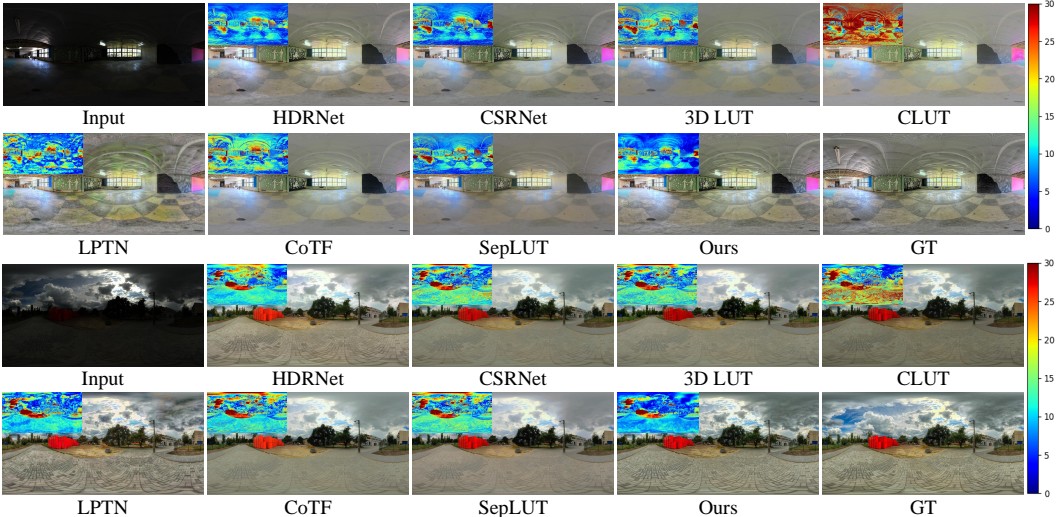

Figure 13: Visual comparisons between our DPRNet and the state-of-the-art methods on the HDRI Haven dataset (4K resolution).

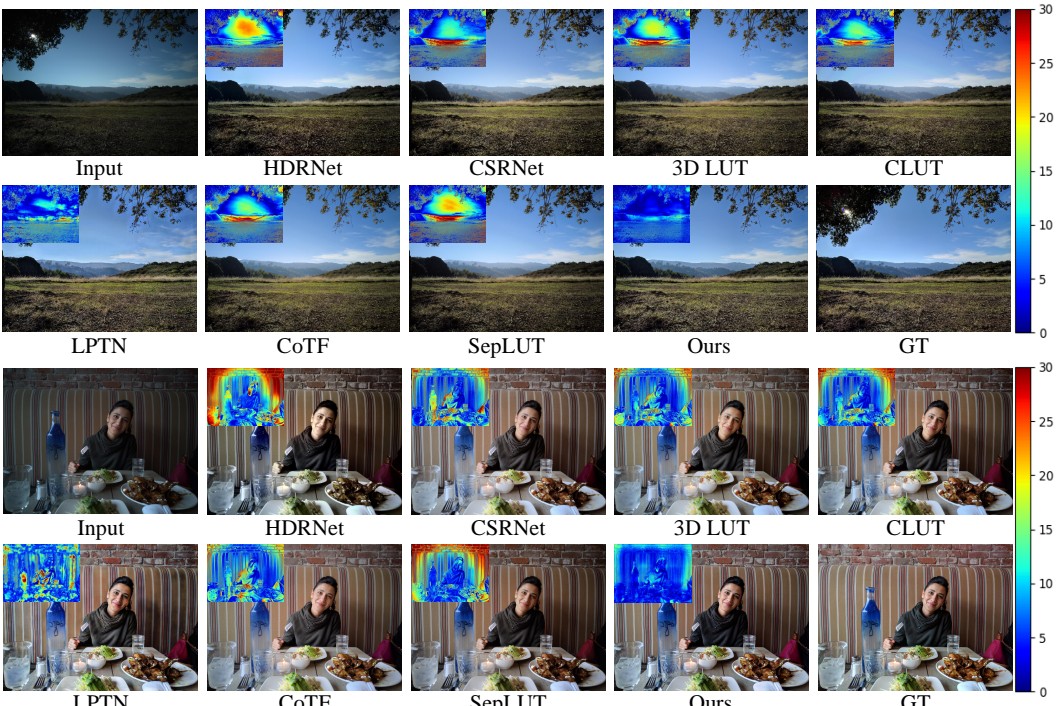

Figure 14: Visual comparisons between our DPRNet and the state-of-the-art methods on the HDR+ dataset (4K resolution).

