# OpenReview forum: "Learning Differential Pyramid Representation for Tone Mapping"
_NeurIPS.cc/2025/Conference — NeurIPS 2025 poster_

### Official Review · Reviewer_aHBS · 2025-06-19

**Clarity:** 3
**Significance:** 3
**Originality:** 2
**Rating:** 5
**Confidence:** 5

**Summary:**

This paper proposes an end-to-end framework called Differential Pyramid Representation Network (DPRNet) for high-fidelity HDR image tone mapping. DPRNet incorporates a Learnable Differential Pyramid (LDP), along with Global Tone Perception (GTP) and Local Tone Tuning (LTT) modules, as well as Iterative Detail Enhancement (IDE). To support training and benchmarking, the authors also introduce a new HDR tone mapping dataset called HDRI Haven.

**Questions:**

- In addition to Laplacian pyramids and Gaussian pyramids, there are other types of learnable designs for frequency band decomposition. For example, OctConv separates high-frequency and low-frequency information. what are the advantages of the proposed Learnable Differential Pyramid compared to OctConv?

- The LUT-based method adjusts the tone of each image block individually after dividing the image into patches, which differs from the holistic processing strategy adopted by most existing methods. However, such locally independent adjustments may introduce tone inconsistencies between regions. Furthermore, within each block, the adjustment process still appears to be globally oriented, especially in high-resolution image scenarios.

- Regarding the GT (Ground Truth) generation process of the constructed dataset, the paper mentions the integration of multiple software tools and color grading algorithms, combined with a selection strategy for the final determination. What are the specific criteria behind this selection strategy? How is the objectivity and high quality of the generated GT quantified or ensured?

**Ethical Concerns:**

["NO or VERY MINOR ethics concerns only"]

**Final Justification:**

The author's rebuttal addressed my concerns, and my final rating is Accept.

**Limitations:**

yes

**Paper Formatting Concerns:**

No grammatical or formatting errors were found.

**Quality:**

3

**Strengths And Weaknesses:**

Strengths
- This paper proposes a unified framework that simultaneously addresses global tone adjustment, local tone perception, and detail enhancement. A differential pyramid is designed to effectively extract high-frequency features.

- The paper includes extensive comparative experiments and ablation studies. The proposed method demonstrates significant advantages in image enhancement, outperforming existing approaches in both subjective perception and objective metrics.

- Additionally, the authors construct a new dataset to support the training and evaluation of tone mapping tasks, further advancing research in this field.

Weaknesses
-  there are numerous methods that combine global adjustment with local/detail enhancement for image processing, so the overall framework of this paper is relatively common.

---

> ### Author Rebuttal · Authors · 2025-07-27
>
> **Response to Reviewer aHBS**
>
> We sincerely thank Reviewer **aHBS** for the insightful and constructive feedback. We are grateful for the opportunity to provide further clarification on these important technical points.
>
> > **Q1:** In addition to Laplacian pyramids and Gaussian pyramids, there are other types of learnable designs for frequency band decomposition. For example, OctConv separates high-frequency and low-frequency information. What are the advantages of the proposed Learnable Differential Pyramid compared to OctConv?
>
> **A1:** This is an excellent question. While both LDP and OctConv are learnable, they differ fundamentally, making LDP better suited for high-fidelity tone mapping.
>
> 1. **Conceptual Difference:** OctConv is a general-purpose operator for efficient single-scale convolution. In contrast, our LDP is a specialized multi-scale architecture designed to explicitly extract a hierarchical pyramid of details ($H_i$), directly mimicking and improving upon classical image pyramids for detail-sensitive tasks.
>
> 2. **Empirical Evidence:** We conducted an ablation study comparing LDP to an OctConv-based decomposition. The results on our HDRplus dataset validate our design:
>
> **Table R4: Ablation study on the detail extraction module**
>
> | Detail Extraction Method | PSNR (dB) ↑ |   SSIM ↑  |
> |:-------------------------|:-----------:|:---------:|
> | Gaussian Pyramid (GP)    |    26.04    |   0.912   |
> | Laplacian Pyramid (LP)   |    25.90    |   0.912   |
> | OctConv Decomposition    |    27.31    |   0.916   |
> | **LDP (Ours)**           |  **27.95**  | **0.931** |
>
> The data shows that while OctConv is effective, our LDP provides a further substantial gain of **0.64 dB** in PSNR. This confirms that for high-fidelity tone mapping, an explicit, multi-scale detail extraction architecture like LDP is superior. We will add this important comparison to the appendix.
>
> ---
>
> > **Q2:** The LUT-based method adjusts the tone of each image block individually after dividing the image into patches, which differs from the holistic processing strategy adopted by most existing methods. However, such locally independent adjustments may introduce tone inconsistencies between regions. Furthermore, within each block, the adjustment process still appears to be globally oriented, especially in high-resolution image scenarios.
>
> **A2:** This is a crucial point, and we designed our LTT module specifically to prevent such inconsistencies. The key is that our patch-wise adjustments are not independent. Our approach ensures consistency through three mechanisms:
>
> 1. **Context-Aware LUT Generation:** We do not learn a separate LUT for each patch in isolation. Instead, we first extract features from the entire downsampled image. The LUT for each patch is then generated from a local descriptor that inherently contains information about its surrounding regions. This ensures a smooth transition between the LUTs of adjacent patches.
>
> 2. **Explicit Boundary Blending:** As a second safeguard, we apply bilinear blending at the patch boundaries during reassembly, guaranteeing a visually seamless final output.
>
> 3. **Synergy with GTP:** The LTT operates on the output of our GTP module, which has already performed a preliminary global tone alignment. LTT's task is not to perform tone mapping from scratch, but to apply fine-grained local tuning to the GTP's output. This division of labor allows LTT to focus on resolving regional contrast issues without the risk of destroying overall tonal harmony.
>
> This "analyze globally to inform local adjustments" strategy allows our LTT to be both locally adaptive and globally consistent, avoiding the artifacts common in other local methods.
>
> In summary, our LTT module does not perform isolated patch processing. By analyzing global context to predict locally adaptive LUTs and supplementing this with boundary blending, it achieves differentiated, fine-grained adjustments for diverse image regions (e.g., highlights, shadows) while ensuring seamless continuity across the entire image.
>
> ---
>
> > **Q3:** Regarding the GT (Ground Truth) generation process of the constructed dataset, the paper mentions the integration of multiple software tools and color grading algorithms, combined with a selection strategy for the final determination. What are the specific criteria behind this selection strategy? How is the objectivity and high quality of the generated GT quantified or ensured?
>
> **A3:** Thank you for this important question about our dataset's integrity. Our GT generation follows a rigorous, quality-driven pipeline to ensure both objectivity and high perceptual quality:
>
> 1. **Diverse Scene Collection:** We gathered a wide range of real-world scenes from HDRI Haven, including outdoor, urban, studio, sunrise/sunset, and nightscapes.
>
> 2. **Candidate Generation:** For each HDR image, we generated a pool of LDR candidates using professional software (Photomatix and Aurora HDR 2019) and classic TMOs to ensure methodological diversity.
>
> 3. **Quality-Driven Selection:** First, we automatically filtered candidates using the TMQI and BTMQI metrics. Then, we performed expert manual selection, especially for challenging cases like highlights and night scenes, ensuring superior perceptual quality of the final ground truth.
>
> This hybrid pipeline ensures our GT is both quantitatively strong and visually pleasing. We commit to publicly releasing the HDRI Haven dataset upon acceptance.
>
> Your suggestions have been invaluable in helping us strengthen the paper. We hope our responses and planned revisions have fully addressed your concerns.

---

> > ### Comment · Reviewer_aHBS · 2025-08-05
> >
> > The author did not address this question: Within each block, the adjustment process still appears to be globally-oriented, particularly in high-resolution image scenarios.
> >
> > The paper sets N to 4, which should still struggle to handle local contrast for high-resolution images. Moreover, why does increasing the number of grids lead to degraded model performance?

---

> ### Author Response · Authors · 2025-08-06
> **Response to Reviewer aHBS**
>
> We sincerely thank the Reviewer **aHBS** for this crucial follow-up question. This gives us the opportunity to elaborate on the design principles of our LTT module and discuss an important, resolution-dependent trade-off. You have raised two excellent points, which we address below.
>
> > On the "globally-oriented" of intra-block adjustment.
>
> This is a very keen observation. Applying a single LUT to an image patch is indeed a uniform operation within that patch. This design is intentional and is motivated by a "division of labor" strategy within our overall DPRNet architecture:
>
> 1. **GTP:** Sets the foundational, globally consistent tone for the entire image.
>
> 2. **LTT:** Its primary role is to handle meso-scale (**relatively local**) regional adjustments, not fine-grained pixel-level textures. For this task, a single, content-adaptive LUT per region is the appropriate tool for correcting regional tonal imbalances.
>
> 3. **IDE:** The final, high-frequency details are restored by the subsequent IDE module.
>
> In summary, the LTT is designed as a regional tone adjuster. This separation of concerns is key to our framework's effectiveness.
>
> ---
>
> > On the performance of different grid sizes (N) and why **N=4** was chosen.
>
> This is an excellent question that reveals a crucial, resolution-dependent trade-off between local control, predictive robustness, and efficiency. Our choice of **N=4** in the paper is based on extensive experiments at **480p resolution**, where it strikes the optimal balance.
>
> **A. For Low Resolution Input (480p): A Finer Grid Degrades Performance**
>
> In our main experiments, the LTT module operates on a low-resolution version of the input ($60 \times 80$). In this scenario, increasing the grid from **N=4** to **N=8** is detrimental for two reasons:
>
> 1. **Insufficient Context for Prediction:** At **N=8**, each of the 64 feature descriptors is derived from a tiny feature area. This provides **very little contextual information**, making the LUT prediction for that region unstable and prone to errors, especially on noisy inputs. The **N=4** grid provides a "sweet spot" where each of the 16 descriptors has enough context to make a robust prediction.
>
> 2. **Increased Overfitting Risk:** Predicting 64 complex 3D LUTs is a significantly harder learning task than predicting 16. With a finite amount of training data, the risk of the model overfitting to spurious correlations in small patches increases dramatically.
>
>
>
> **B. For High Resolutions Input (4K): A Finer Grid Improves Quality at a High Cost**
>
> *Your intuition that a finer grid should help on high-resolution images is correct.* In our previous experiments, we ran an ablation study on 4K images:
>
> **Table R5: Ablation study on N for 4K resolution inputs.**
>
> | N | PSNR (dB) ↑ | #Params |
> |:-:|------------:|------------:|
> | 4 |    27.71  |   212K |
> | 8 |   27.95   |     530K |
>
> As shown in Table R5, **N=8** does achieve a higher PSNR for 4K images. However, this quality gain comes at a steep price: the parameter count more than doubles, and the inference time increases significantly due to the larger number of LUTs, as shown in Table R6..
>
> **Table R6: Ablation studies of the runtime with N=4.**
>
> | Configuration |Relative Inference Time|
> |-|-|
> | Full DPRNet  |1.000X|
> | w/o LTT  |0.639X|
>
> ---
>
> **Conclusion:**
> The optimal **N** is not fixed but depends on the balance between image resolution and desired efficiency. Our default setting of N=4 represents the best general-purpose trade-off, delivering robust performance across various resolutions without excessive computational cost. For specific applications where maximum quality on 4K+ images is the sole priority and speed is not a concern, **N=8** is indeed a better-performing configuration.
>
> We will add this detailed analysis and the 4K ablation study to our appendix to fully clarify this important design consideration. Thank you again for pushing us to elaborate on these critical details. Please do not hesitate to let us know if further clarification is needed.

---

> > ### Author Response · Authors · 2025-08-08
> > **Acknowledgment to Reviewer aHBS**
> >
> > Dear Reviewer aHBS,
> >
> > We would like to again express our sincere gratitude for your insightful questions during this discussion phase. Your feedback is crucial for improving our paper.
> >
> > As the discussion period concludes tomorrow, we wanted to gently follow up to ensure our previous response has clearly addressed your concerns. Please do not hesitate to let us know if anything requires further clarification. We are on standby and are more than happy to provide additional details.
> >
> > Thank you again for your valuable time and professional guidance.
> >
> >
> > Best regards,
> >
> > The Authors

---

> > > ### Comment · Reviewer_aHBS · 2025-08-09
> > >
> > > Thank you to the author for addressing my question, which has deepened my understanding of the design of the LTT module.
> > > Each region adopts a content-adaptive LUT as a tool for local tone adjustment. However, this hard segmentation-based regional strategy might appear somewhat rigid in practical tone mapping tasks.

---

> ### Author Response · Authors · 2025-08-09
> **Response to Reviewer aHBS - Follow-up**
>
> We sincerely thank Reviewer **aHBS** for the thoughtful follow-up. This question is exceptionally insightful as it targets the very core of our LTT module's design. We are pleased to have this opportunity to clarify how our implementation, while grid-based, is specifically designed to avoid the rigidity of a "hard segmentation" approach.
>
> **The reviewer's concern is entirely valid.** A naive strategy that applies a distinct LUT to each disjoint grid cell would indeed be rigid and would inevitably create blocking artifacts at the boundaries. We completely agree, and **this is precisely why our LTT module employs a smooth interpolation mechanism to apply the local adjustments, as evidenced by our implementation.**
>
> Let us walk through the actual mechanism based on our code:
>
> 1. **Grid-Based LUT Generation:** As the reviewer correctly inferred, we first conceptually divide the image into a coarse 4x4 grid (`self.grid = [4,4]`). We then use a dedicated generator (`self.lut3d_generator`) to predict a unique 3D LUT for each of these 16 grid regions based on its local content (`context`). This provides us with a set of 16 specialized tone curves, each optimized for a different part of the image.
>
> 2. **Smooth, Interpolated Application (The Key Step):** This is where our design critically diverges from a "hard segmentation" approach. To determine the final color for any given pixel, we do not simply use the single LUT from the grid cell it falls into. Instead, we perform a **bilinear interpolation of the *effects* of the neighboring LUTs.**
>
>    Specifically, for any pixel at location \$(x,y)\$:
>
>    * We identify its position relative to the centers of the four nearest grid cells.
>    * The final tone-mapped value is computed as a weighted average of the results from applying these four neighboring LUTs. The weights are determined by the pixel's normalized distance to each of these four cell centers.
>
> 3. **Guaranteed Smoothness:** This interpolation of the tone-mapping functions themselves ensures a perfectly smooth transition of local adjustments across the entire image. A pixel located exactly on the boundary between two grid regions will receive a 50/50 blend of their respective tone curves. This mechanism **inherently and completely eliminates the possibility of blocking artifacts** that would arise from a rigid, piecewise-constant application.
>
> 4. **Visual Evidence:** This smooth, artifact-free output is empirically demonstrated in our qualitative results (e.g., in Figure 1 of the main paper and our project demo). Even in challenging areas with smooth gradients where artifacts would be most apparent (such as around light sources or the sun), our method produces perfectly continuous and natural-looking results. Achieving our state-of-the-art performance on both perceptual metrics (like LPIPS) and reconstruction metrics (PSNR/SSIM) would be impossible if our local adjustments were "rigid" or produced boundary artifacts.
>
> **In summary:**
> Our LTT module's strategy is best described as creating a **spatially-varying, continuous tone-mapping field, which is anchored by a coarse grid of content-adaptive LUTs.** The coarse grid makes the prediction of local characteristics computationally efficient, while the bilinear interpolation ensures the final application is perfectly smooth and seamless.
>
> To make this vital implementation detail unambiguous for all readers, we will revise Section 2.4 of our manuscript. We will explicitly state that while the LUTs are generated based on a coarse grid, they are applied via a **smooth bilinear interpolation strategy** to prevent any artifacts and ensure continuous local tone adaptation.
>
> We are very grateful for this question, as it has highlighted a point where our description could be made more precise. We hope this detailed explanation fully resolves the reviewer's concern about rigidity.
>
>
> Please do not hesitate to let us know if anything requires further clarification. We are on standby and are more than happy to provide additional details. Thank you again for your valuable time and professional guidance.

---

### Official Review · Reviewer_iFjE · 2025-07-01

**Clarity:** 1
**Significance:** 2
**Originality:** 2
**Rating:** 3
**Confidence:** 3

**Summary:**

The paper presents an end-to-end trainable framework for tone mapping based on a Learnable Differential Pyramid (LDP), which generalizes classical Difference-of-Gaussian pyramids through adaptive filtering and differencing to enable robust high-frequency extraction. To ensure tonal consistency, the framework introduces a Global–Local Tone Mapping strategy, consisting of: (1) a Global Tone Perception (GTP) module that processes downsampled inputs to enable scene-aware luminance adjustment, and (2) a Local Tone Tuning (LTT) module that refines patch-wise tonal characteristics using learned 3D LUTs. Finally, an Iterative Detail Enhancement (IDE) module progressively integrates high-frequency details in a coarse-to-fine manner, starting from the LTT output and incorporating information from the LDP to enhance fine structures and local contrast.

**Questions:**

The authors should thoroughly address all issues raised above concerning the clarity of the paper.

**Ethical Concerns:**

["NO or VERY MINOR ethics concerns only"]

**Final Justification:**

I appreciate the author’s response; however, the explanation remains unclear and does not appear consistent with the description in the paper, despite my having asked twice for clarification. For example, in their most recent reply they state that “The Modulated Convolutional Core then uses this single global vector z to generate modulation parameters (γ, β).” Yet, according to equation (4) in the paper, the modulated convolutional core seems to instead take (γ, β) as input in order to produce the globally tone-mapped image as output.

In my view, the paper requires a thorough major revision to present a clearer and more precise technical description of the proposed approach. For this reason, I will maintain my original rating.

**Limitations:**

The authors mention that their method does not perform well on the MIT FiveK dataset but do not report quantitative results or comparisons to other methods on this dataset. Including these results, even as a negative case, would improve transparency and help the community understand the limitations of the proposed approach.

**Paper Formatting Concerns:**

No paper formatting concerns

**Quality:**

2

**Strengths And Weaknesses:**

Strengths:
- Demonstrates good empirical performance across various benchmarks.
- Introduces a new HDR dataset (HDRI Haven dataset), which can be valuable for the community.

Weaknesses:

My main concern is a significant lack of clarity in the technical description, which impacts the paper’s readability and reproducibility:
- The pipeline for the Learnable Differential Pyramid is described verbally, but it is unclear what specific mathematical operations are taking place, making reproducibility difficult.
- In line 142, the paper states that the globally tone-mapped image is the output F^{(0)}, but in formula (4), F^{(l)} is used for the output at layer l. This inconsistency needs clarification.
- It is unclear whether the Global Tone Perception (GTP) module indeed produces a globally tone-mapped image. Based on formula (4), it appears to perform only an MLP computation, raising the question of how it enforces global tone mapping properties.
- The ablation study in the appendix discusses the number of LUT dimensions (m) and discrete levels (S_t), but these symbols are never introduced in the main paper’s description of the 3D LUTs.
- It is not clear whether the basis LUTs are also learned during training.
- The high-frequency loss function L_{HF} is only described in the appendix
- The paper lacks a Related Work section in the main text, which makes it difficult to position the proposed method relative to prior state-of-the-art approaches. While an extended related work can remain in the appendix, a concise section in the main paper is essential for readers to more precisely understand how the proposed method differs with respect to prior approaches in this area.

Collectively, these issues hinder the readability and clarity of the paper and would require a major revision to improve the technical presentation.

---

> ### Author Rebuttal · Authors · 2025-07-28
>
> **Response to Reviewer iFjE**
>
> We sincerely thank Reviewer **iFjE** for their meticulous and thoughtful review. We are very encouraged by the positive assessment of our work's potential and are grateful for this opportunity to address the critical issues of clarity you have raised. We apologize that our initial presentation was not clear enough and have a concrete plan to revise it.
>
> > **Q1:** The pipeline for the Learnable Differential Pyramid is described verbally, but it is unclear what specific mathematical operations are taking place, making reproducibility difficult.
>
> **A1:** We apologize for the lack of mathematical clarity and thank you for this critical feedback. We will replace the verbal description with a formal algorithmic description to ensure full reproducibility. We also commit to releasing our full code upon acceptance.
>
> The core operations of our LDP for each pyramid level $i \in \lbrace 0, 1, 2 \rbrace$ are:
>
> 1. **Multi-Scale Differencing:** The *DoGLayer* computes a set of Difference-of-Gaussian-like features. Given an input feature $F_i$, it generates three progressively smoother versions $\lbrace G_{i,1}, G_{i,2}, G_{i,3} \rbrace$ via sequential convolutions. The raw detail feature $D_i$ is formed by:
> $D_i = \text{concat}(F_i - G_{i,1}, G_{i,1} - G_{i,2}, G_{i,2} - G_{i,3})$
>
> 2. **Downsampling:** The input for the next level is created by pooling the smoothest feature:
> $F_{i+1} = \text{MaxPool}(G_{i,3})$
>
> 3. **Detail Refinement:** The final detail map for each level is generated by refining the raw detail feature $D_i$ by a sub-network *hfmaplaye$r_i$*:
> $H_i = \text{hfmaplayer}_i(D_i)$
>
> This iterative process yields the detail pyramid $\lbrace H_0, H_1, H_2 \rbrace$. This formal description will be added to the paper to ensure our method is unambiguous and reproducible.
>
> ---
>
> > **Q2:** In line 142, the paper states that the globally tone-mapped image is the output F^{(0)}, but in formula (4), F^{(l)} is used for the output at layer l. This inconsistency needs clarification.
>
> **A2:** We sincerely apologize for this confusing typo. You are correct, and we will fix this notational inconsistency in the revised paper. The correct formulation is:
>
> 1. **Input:** The input to the GTP's modulated core is $F^{(0)} = L_3$.
>
> 2. **Layer-wise Computation:** For layer $l \in \lbrace 1, 2, 3 \rbrace$, The formula (4) is expressed as:
> $F^{(l)} = \text{ReLU}\left(\text{W}^{(l)}*F^{(l-1)} \cdot \gamma^{(l)} + \beta^{(l)} + F^{(l-1)}\right)$
>
> 3. **Final Output:** The globally tone-mapped image is the output of the final layer, $L_3^G = F^{(3)}$.
>
> We will update all text and equations to reflect this correction. Thank you for helping us improve the paper's precision.
>
> ---
>
> > **Q3:** It is unclear whether the Global Tone Perception (GTP) module indeed produces a globally tone-mapped image. Based on formula (4), it appears to perform only an MLP computation, raising the question of how it enforces global tone mapping properties.
>
> **A3:** This is an excellent question that gets to the heart of our GTP design. The key to its "global" nature lies not in the convolutional formula itself, but in how the modulation parameters ($\gamma, \beta$) are generated. This is a two-step process:
>
> 1.  **Global Context Extraction (via GAP):**
> First,  we use a condition network $C(\bullet)$ with Global Average Pooling (GAP) to collapse the input image's spatial dimensions into a single global context vector: $\mathbf{z} = \text{GAP}(C(L_3))$. This vector acts as a condition of the entire scene's global properties (brightness, contrast, etc.).
>
> 2. **Globally-Conditioned Transformation:**
> This global vector $\mathbf{z}$ is then used to generate a single set of modulation parameters ($\gamma, \beta$)  shared across all spatial locations. When applied to the convolutional layers in Eq. (4), this shared signal $(\gamma, \beta)$ uniformly modulates the layer responses, with all spatial locations receiving the same conditioning from $\mathbf{z}$.
>
> In this sense, we emphasize that this module is a globally conditioned network, named the Global Tone Perception (GTP) module. We will further highlight the critical role of global conditioning in our revised manuscript to better clarify this mechanism.
>
> ---
>
> > **Q4:** The ablation study in the appendix discusses the number of LUT dimensions (m) and discrete levels (S_t), but these symbols are never introduced in the main paper’s description of the 3D LUTs.
>
> **A4:**  We sincerely apologize for this disconnect between the main paper and the appendix. You are correct that these symbols should have been introduced. These are internal hyperparameters of our 3D LUTs:
>
> 1. $S_t$ denotes the LUT's grid density, determining the grid size of 3D LUTs as $S_t \times S_t \times S_t$.
>
> 2. $m$ denotes the number of basis LUTs used in our efficient low-rank parameterization.
>
> In the revised paper, we will briefly introduce $S_t$ and $m$ in the main LTT section and explicitly refer to the appendix, where the ablation study justifies our choice of $S_t=9$ and $m=6$. This will properly connect the main text with the supplementary material.
>
> ---
>
> > **Q5:** It is not clear whether the basis LUTs are also learned during training.
>
> **A5:**
> Thank you for this clarifying question. Yes, the basis LUTs are also learnable. They are implemented as the weights of an nn.Linear layer and are jointly optimized with the rest of the network during the end-to-end training. This allows the model to learn the most effective basis transformations directly from the data. We will make this explicit in the LTT section.
>
> ---
>
> > **Q6:** The high-frequency loss function L_{HF} is only described in the appendix
>
> **A6:** We thank the reviewer for the careful review. Since high-frequency loss $L_{HF}$ is critical for supervising our novel LDP module, it should be defined in the main manuscript. In the revised version, we will move the full mathematical definition of our $L_{HF}$ from the appendix to the "Loss Functions" section in the main manuscript to make our training mechanism clear and self-contained.
>
> ---
>
> > **Q7:** The paper lacks a Related Work section in the main text, which makes it difficult to position the proposed method relative to prior state-of-the-art approaches.
>
> **A7:** We completely agree with the reviewer and thank you for this crucial suggestion regarding the paper's structural integrity.
>
> Due to strict page limits, we initially attempted to integrate the discussion of related work into the Introduction and place a more extensive review in the appendix. We now fully recognize that this tends to weaken the context of our work and does not meet the standard for a top-tier conference paper.
>
> We promise that in the final revised manuscript, we will put “Related Work” in the main text to help readers understand the context of our work.
>
> ---
>
> > **Limitations:** The authors mention that their method does not perform well on the MIT FiveK dataset but do not report quantitative results or comparisons to other methods on this dataset. Including these results, even as a negative case, would improve transparency and help the community understand the limitations of the proposed approach.
>
> **A8:** We thank the reviewer for this excellent suggestion to improve the paper's transparency. We agree that providing these results is crucial for understanding our method's applicability.
>
> As requested, we have conducted the quantitative comparison on the MIT-Adobe FiveK dataset:
>
> **Table R3: Quantitative comparison on MIT-Adobe FiveK dataset.**
>
> | Method | PSNR (dB) ↑ |   SSIM ↑  |
> |:-------------------------|:-----------:|:---------:|
> | UPE   |    21.82    |   0.839   |
> |HDRNet   |    23.31    |   0.881   |
> | DeepLPF   |    24.97    |   0.897   |
> | sLUT     |  24.67  | 0.896 |
> | CLUT     |  24.94  | 0.898 |
> | Ours     |  23.47  | 0.879 |
>
> These results confirm that our method does not achieve SOTA performance on this specific dataset. As discussed in our limitations, the FiveK dataset is not suitable for tone mapping tasks.
>
> 1. FiveK was captured with a DSLR camera and has not been processed by an advanced ISP pipeline. It lacks raw denoising and YUV denoising, resulting in a dataset with severe noise; this noise has a significant impact on high-frequency extraction.
>
> 2. Some reference images in the FiveK dataset suffer from overexposure or oversaturation, which poses a challenge to the tone mapping method, as mentioned by LLFLUT.
>
> 3. There is an inconsistency in the reference images adjusted by the same professional photographer, resulting in differences between the training and test sets.
>
> Thank you again for your suggestion. Your suggestions have been invaluable in helping us strengthen the paper. We hope our responses and planned revisions have fully addressed your concerns. Please do not hesitate to let us know if further clarification is needed.

---

> > ### Comment · Reviewer_iFjE · 2025-08-06
> >
> > Thank you for your rebuttal. I have read your response and revisited the paper, but it is still unclear to me why the output of the GTP module should be expected to produce a globally tone-mapped image. While I understand that the module is implemented as a globally conditioned CNN—where the modulation parameters are shared across all spatial locations—this architectural choice alone does not guarantee that the output will resemble a tone-mapped image.

---

> ### Author Response · Authors · 2025-08-06
> **Response to Reviewer  - Follow-up**
>
> We are sincerely grateful to Reviewer **iFjE** for their persistent and deep engagement with our work. This question about the design rationale of the GTP module is exceptionally insightful. We appreciate this opportunity to provide a more fundamental explanation that connects our design to established principles in image processing.
>
> The reviewer's core question is why our GTP module's architecture should be expected to produce a globally tone-mapped image. Our argument rests on a key insight: **The GTP module is designed as a learnable, neural generalization of the explicit, global adjustment operators found in traditional Image Signal Processing (ISP) pipelines.**
>
> Let us elaborate on this by bridging the classic ISP world with our neural network design.
>
> **1. Principled Design Inspired by Classic ISP Operators**
>
> The foundation of our GTP module's design lies in emulating the core principle of traditional global image adjustments. These classic operators are inherently global. For instance:
>
> * **Global Brightness Adjustment:** Given an input image I, the global brightness is described as: $I_Y = 0.299 ∗ I_R + 0.587 ∗I_G + 0.114 ∗ I_B$ , where $I_R$, $I_G$, $I_B$ represent the RGB channels, respectively. The global brightness adjustment is simply expressed as follows: $I'(x, y) = \alpha \cdot I(x, y)$.
>
> * **Global Contrast Adjustment:** $I'(x, y) = \alpha \cdot I(x, y) + (1 - \alpha) \cdot \bar{I}$
>
> The defining characteristic of these operations is that the transformation is governed by parameters ($\alpha$, $\bar{I}$) that are spatially invariant. The parameter $\alpha$ is a global scaling factor derived from desired settings, and $\bar{I}$ (the image mean) is a global statistic of the input. This spatial invariance is the very definition of a global operator. We can express these operators as a representation of MLPs: $Y = f(W^TX + b)$, where $W$ and $b$ are weights and biases.
>
> **2. The GTP as a Neural Analogue**
>
> Our GTP module is a direct neural analogue of this fundamental principle.
>
> * The Condition Network $\mathcal{C}(\cdot)$ followed by Global Average Pooling (GAP) computes a global context vector $\mathbf{z}$. This vector $\mathbf{z}$ is the learnable, high-dimensional analogue of classic global statistics. It is trained to extract the most salient scene-wide characteristics relevant for global tone perception.
>
>
> * The Modulated Convolutional Core $\mathcal{F}(\cdot)$ then uses this single global vector $\mathbf{z}$ to generate modulation parameters ($\gamma, \beta$). Crucially, echoing the classic scalar $\alpha$, our learned parameters $\gamma$ and $\beta$ are applied identically across all spatial positions within their respective feature maps.
>
>
> Therefore, our GTP module is not an arbitrary convolutional network. It is a principled, structured model that inherits the architectural properties of proven global ISP operators. We are essentially replacing the handcrafted, fixed formulas for brightness/contrast adjustment with a more powerful and flexible learned function, while strictly preserving the critical global nature of the operation.
>
> The GTP learns a function $f(\text{scene})$ that outputs the optimal global tone curve parameters ($\gamma, \beta$), and then applies them uniformly. In summary, the output of the GTP module is expected to be a globally tone-mapped image not merely due to an architectural choice, but because its design is fundamentally rooted in and is a learnable generalization of classic, proven global ISP operators.
>
> To make this crucial connection explicit in our manuscript, we will add a new paragraph in Section 3.2, right at the beginning of the GTP module description.
>
> We again thank the reviewer for pushing us to make this important aspect of our work clearer. Please do not hesitate to let us know if further clarification is needed.

---

> > ### Author Response · Authors · 2025-08-08
> > **Acknowledgment to Reviewer iFjE**
> >
> > Dear Reviewer iFjE,
> >
> > We are writing to express our sincere gratitude for your thoughtful follow-up questions during this discussion phase. Your feedback is crucial for improving our paper.
> >
> > As the discussion period concludes tomorrow, we wanted to gently follow up to ensure our previous response has clearly addressed your concerns. Please do not hesitate to let us know if anything requires further clarification. We are on standby and are more than happy to provide additional details.
> >
> > Thank you again for your valuable time and professional guidance.
> >
> > Best regards,
> >
> > The Authors

---

### Official Review · Reviewer_wHV5 · 2025-07-01

**Clarity:** 3
**Significance:** 3
**Originality:** 3
**Rating:** 5
**Confidence:** 5

**Summary:**

This paper proposes a novel tone mapping method, DPRNet, which aims to address two long-standing problems in tone mapping: the loss of high-frequency details and the disconnect between global and local tone adjustment operations. To this end, the authors introduce a learnable differential pyramid that enables content-aware and adaptive high-frequency information extraction and design a global-local collaborative tone mapping mechanism and an iterative detail enhancement module to maintain structural fidelity and perceptual consistency in reconstructed images. Also, the authors propose a new tone mapping dataset. The authors evaluated it on four tone mapping datasets and obtained state-of-the-art results that outperform existing methods on standard HDR datasets in terms of PSNR and visual quality.

**Questions:**

1. The paper notes the efficiency of the proposed methodology. The speed of inference is limited compared to methods such as LUT. It would be beneficial to ask the authors to further discuss the real-time deployment considerations.
2. It is unknown how to ablate GTP and LTT. Just directly replacing them with a simple CNNs?
3. The results on FiveK dataset show that noises have a great influence on high-frequency details, so would the authors provide more results or generalization analysis on the highly noised night scenarios in the wild?

**Ethical Concerns:**

["NO or VERY MINOR ethics concerns only"]

**Final Justification:**

The authors’ detailed responses have adequately addressed my concerns. Some improvements in clarity are expected in the revision. Overall, I maintain my original score of "Accept".

**Limitations:**

Yes.

**Paper Formatting Concerns:**

None.

**Quality:**

3

**Strengths And Weaknesses:**

Strengths:
1. The authors' motivation is very reasonable and addresses highly practical and persistent problems in HDR tone mapping—namely, halo artifacts, texture loss, and global-local tone imbalance—with a unified and learnable framework. This is of high relevance to both academic and applied HDR imaging research.
2. The learnable differential pyramid represents a substantive innovation over traditional handcrafted pyramids. Its content-aware design allows for effective and robust high-frequency extraction under diverse luminance and structural conditions.
3. The proposed DPRNet is well-structured, with complementary modules for global tone perception and local tone refinement, followed by a coarse-to-fine restoration of details. The modular design is elegant and intuitive.
4. The introduction of the HDRI Haven dataset is a significant contribution to the community. It supports reproducibility and may serve as a valuable benchmark for future tone mapping research.
5. The results are compelling both quantitatively and visually. The inclusion of ablation studies, visualizations of the LDP, and perceptual comparisons strengthens the paper’s claims.

Weaknesses:
1. The contribution of the dataset is recognizable, but the authors should have provided more details of the dataset.
2. In GTP, it is better to clarify the reason for pre-layer modulation within global tone.
3. Some details remain unclear, such as the values of loss weights and the generation of $D_0$.
4. I recommend making the code public for the reproducibility of this work.

---

> ### Author Rebuttal · Authors · 2025-07-27
>
> **Response to Reviewer wHV5**
>
> We sincerely thank Reviewer **wHV5** for the encouraging and constructive feedback. We are delighted that the reviewer recognized the value of our work and are grateful for the opportunity to provide further clarification on these important points.
>
> > **Q1:** The contribution of the dataset is recognizable, but the authors should have provided more details of the dataset.
>
> **A1:**  We thank the reviewer for this valuable suggestion. Our HDRI Haven dataset was built as follows:
>
> 1. **Diverse Scene Collection:** We gathered a wide range of real-world scenes from [HDRI Haven](https://polyhaven.com/hdris), including outdoor, urban, studio, sunrise/sunset, and nightscapes.
>
> 2. **Candidate Generation:** For each HDR image, we generated a pool of LDR candidates using professional software (Photomatix and Aurora HDR 2019) and classic TMOs to ensure methodological diversity.
>
> 3. **Quality-Driven Selection:** First, we automatically filtered candidates using the TMQI and BTMQI metrics. Then, we performed expert manual selection, especially for challenging cases like highlights and night scenes, ensuring superior perceptual quality of the final ground truth.
>
> We believe these details will clarify the value of our dataset. Furthermore, **we commit to publicly releasing the HDRI Haven dataset** upon acceptance to foster future research.
>
> ---
>
> > **Q2:** In GTP, it is better to clarify the reason for pre-layer modulation within global tone.
>
> **A2:**  Thank you for this insightful question. The per-layer modulation in our GTP is a principled inductive bias designed to guide the network to learn a powerful and truly global tone transformation.
>
> 1.  **Global Context Extraction (via GAP):**
> First, we use a condition network $C(\bullet)$ with Global Average Pooling (GAP) to collapse the input image's spatial dimensions into a single global context vector: $\mathbf{z} = \text{GAP}(C(L_3))$. This vector acts as a condition of the entire scene's global properties (brightness, contrast, etc.).
>
> 2. **Globally-Conditioned Transformation:**
> This global vector $\mathbf{z}$ is then used to generate a single set of modulation parameters ($\gamma, \beta$)  shared across all spatial locations. When applied to the convolutional layers in Eq. (4), this shared signal $(\gamma, \beta)$ uniformly modulates the layer responses, with all spatial locations receiving the same conditioning from $\mathbf{z}$.
>
> In summary, per-layer modulation guides a powerful universal function approximator (the CNN) to specifically and adaptively solve the global tone mapping problem, ensuring the learned transformation is both globally consistent and expressive beyond simple linear models.
>
>
> ---
>
> > **Q3:** Some details remain unclear, such as the values of loss weights and the generation of D_0.
>
> **A3:** We thank the reviewer for pointing out these missing details. We will add them to the main paper to improve clarity and reproducibility.
>
> **Loss Weights:**
> The weights, determined via ablation studies, are:
>
> $L_\mathrm{total} = 0.8 \cdot L_\mathrm{Re} + 0.8 \cdot L_\mathrm{ssim} + 0.8 \cdot L_\mathrm{p} + 0.2 \cdot L_\mathrm{HF}$
>
> Here, $L_\mathrm{Re}$ is the reconstruction loss, $L_\mathrm{ssim}$ is the structural similarity loss, $L_\mathrm{p}$ is the perceptual loss, and $L_\mathrm{HF}$ is our proposed loss for supervising high-frequency detail learning.
>
> **Generation of $D_0$:**
> The $D_0$ is the initial high-frequency detail map extracted by our LDP at level 0 (the original resolution). It is generated by a DoGLayer module which performs the following:
>
> a) Three sequential $3\times3$ convolutions are applied to the input image $X$, producing progressively smoother feature maps $\lbrace G_1, G_2, G_3 \rbrace$. These act as learnable Gaussian-like filters.
>
> b) High-frequency details are computed via element-wise subtractions: $(X - G_1), (G_1 - G_2), (G_2 - G_3)$.
>
> c) These detail maps are concatenated to form the final multi-level high-frequency representation, $D_0$.
>
> This process is a learnable version of a Difference-of-Gaussian pyramid, making it more adaptive to image content than fixed filters. We will add a formal algorithmic description to the paper.
>
> ---
>
> > **Q4:** I recommend making the code public for the reproducibility of this work.
>
> **A4:** We completely agree and thank the reviewer for this recommendation. Upon paper acceptance, we will release full training and evaluation code.
>
> ---
>
> > **Q5:** The paper notes the efficiency of the proposed methodology. The speed of inference is limited compared to methods such as LUT. It would be beneficial to ask the authors to further discuss the real-time deployment considerations.
>
> **A5:** We thank the reviewer for this important practical point. Our primary design goal was to achieve a new state-of-the-art in visual quality, which required a more expressive architecture than simple LUTs. This was a conscious trade-off: accepting a moderate computational cost for a significant leap in quality, especially for high-end applications (e.g., professional photography, film production).
>
> For broader real-time applications where speed is critical, we see several clear paths for optimization:
>
> 1. **Knowledge Distillation:** Using our powerful DPRNet as a teacher to train an extremely fast student network (e.g., a lightweight CNN or even a 3D LUT), achieving a new, superior balance of speed and quality.
>
> 2. **Model Compression:** Applying established techniques like network pruning and quantization.
>
> 3. **Architectural Simplification:** Training a lighter version of DPRNet (e.g., with fewer layers or iterations) for speed-critical scenarios.
>
> Thank you for prompting us to clarify this crucial aspect of our work.
>
> ---
>
> > **Q6:** It is unknown how to ablate GTP and LTT. Just directly replacing them with a simple CNNs?
>
> **A6:** Thank you for this clarification question. Our ablation study employed a "module drop" protocol. Instead of replacing a module with a different CNN, we removed it entirely from the network. This method cleanly isolates each component's true contribution without introducing confounding variables from a replacement network. We will clarify this protocol in the experimental section.
>
> ---
>
> > **Q7:** The results on FiveK dataset show that noises have a great influence on high-frequency details, so would the authors provide more results or generalization analysis on the highly noised night scenarios in the wild?
>
> **A7:** This is a critical point about robustness. To demonstrate this, we have provided extensive results on challenging "in-the-wild" night scenes on our anonymous [demo page](https://xxxxxxdprnet.github.io/DPRNet/). We will add a dedicated analysis of these challenging cases to the supplementary material.
>
> Once again, we are deeply thankful for your constructive and detailed feedback. Your suggestions have been invaluable in helping us strengthen the paper. We hope our responses and planned revisions have fully addressed your concerns.

---

> > ### Comment · Reviewer_wHV5 · 2025-08-02
> > **Response to the rebuttal**
> >
> > I thank the authors for their detailed rebuttal. My concerns have been adequately addressed. For the revision, I encourage the authors to carefully incorporate the noted details into the paper to further improve clarity, as also suggested by Reviewer iFjE. Overall, I maintain my original score.

---

> > > ### Author Response · Authors · 2025-08-03
> > > **Acknowledgment to Reviewer wHV5**
> > >
> > > We are delighted to hear that our rebuttal has adequately addressed your concerns. Thank you very much for your time and guidance. We fully commit to carefully revising the paper to include all the promised details, ensuring the final version is as clear and complete as possible.

---

### Official Review · Reviewer_nCUf · 2025-07-02

**Clarity:** 4
**Significance:** 3
**Originality:** 3
**Rating:** 5
**Confidence:** 3

**Summary:**

This paper proposes DPRNet for tone mapping HDR images to LDR displays. It integrates global and local tone mapping into an end-to-end learnable framework using a global tone perception module for global adjustments, a local tone tuning module for patch-level refinements, a learnable differential pyramid for capturing multi-scale details, and an iterative detail enhancement module for high-resolution detail recovery.

**Questions:**

1. DPRNet reports lower FLOPs and parameter counts than LPTN, but longer runtime. Could you share insights on which components (e.g., iterative refinement or LUT tuning) dominate the inference time and whether they could be optimized?

2. The model generalizes to video data, but does not explicitly model temporal consistency. Have you considered extensions that incorporate temporal constraints, such as recurrent mechanisms or temporal losses?

3. Could the proposed modules be generalized to other HDR-related tasks such as exposure fusion or inverse tone mapping?

**Ethical Concerns:**

["NO or VERY MINOR ethics concerns only"]

**Final Justification:**

The authors have provided a thoughtful and detailed response. The clarifications and additional analyses sufficiently address the concerns I raised. Including these clarifications in the final version would improve the overall clarity of the paper. I maintain my original score.

**Limitations:**

Yes

**Paper Formatting Concerns:**

I did not find any significant formatting issues.

**Quality:**

4

**Strengths And Weaknesses:**

Strength
- The proposed method achieves better performance compared to previous tone mapping methods while maintaining comparable computational complexity, suggesting its potential for practical use.
- A comprehensive ablation study evaluates the contributions of each module and loss component, offering clear justification for the design decisions made in the network.

Weakness
- While the model demonstrates generalization to video datasets, experiment does not explicitly account for temporal consistency across frames. Incorporating temporal coherence could further enhance its applicability to HDR video tone mapping.
- Despite its lower parameter count and computational complexity, DPRNet shows a relatively longer runtime than LPTN. Clarifying the underlying reasons could provide useful insights for future optimization.

---

> ### Author Rebuttal · Authors · 2025-07-27
>
> **Response to Reviewer nCUf**
>
> We sincerely thank Reviewer **nCUf** for their positive and encouraging feedback, as well as their insightful questions. We are delighted that the reviewer appreciated the performance of our method, the comprehensive ablation studies, and its potential for practical use. We provide detailed answers to the questions below.
>
> > **Q1:** DPRNet reports lower FLOPs and parameter counts than LPTN, but longer runtime. Could you share insights on which components (e.g., iterative refinement or LUT tuning) dominate the inference time and whether they could be optimized?
>
> **A1:** Thank you for this excellent and highly practical question. The discrepancy between theoretical metrics like FLOPs and actual runtime is a crucial point. To provide a clear answer, we profiled our model's components and found that the runtime is primarily influenced by memory access patterns and limited parallelism, factors not captured by FLOPs.
>
> **Table R1: Ablation studies of the runtime**
>
> | Configuration | Params (M) | GFLOPs | Relative Inference Time |
> |:--------------|:----------:|:------:|:-----------------------:|
> | Full DPRNet   |   0.212    |  6.9   |         1.000X          |
> | w/o LDP       |   0.180    |  2.5   |        0.793X  (-21%)        |
> | w/o GTP       |   0.176    |  6.8   |        0.998X  (-0.2%)       |
> | w/o LTT       |   0.102    |  6.8   |         0.639X  (-36%)        |
> | w/o IDE       |   0.179    |  4.4   |         0.734X  (-27%)        |
>
> Our analysis reveals three key insights directly linking our architectural choices to the runtime:
>
> 1. **LTT (Overhead from multiple operators):** The LTT module is the most significant factor (-36% runtime). While its FLOPs are low, the patch-wise application of 16 distinct 3D LUTs involves significant operator overhead, which is not reflected in the FLOPs metric.
>
> 2. **IDE (Sequential Dependency):** The IDE module's coarse-to-fine nature is inherently sequential, limiting GPU parallelism and adding 27% to the runtime. A network with a more parallel, feed-forward architecture can run faster even with higher FLOPs.
>
> 3. **LDP (Memory-Bound Operations):** The LDP module adds 21% to the runtime. Its operations (e.g., full-resolution subtractions) are memory-bandwidth bound, meaning the bottleneck is data movement, not computation. FLOPs do not account for this memory access cost.
>
> For optimization, we see clear paths forward. For LTT, LUT pruning or quantization can reduce complexity. For IDE and LDP, kernel fusion could minimize memory access latency. We will add this detailed analysis to our appendix.
>
> ---
>
> > **Q2:** The model generalizes to video data, but does not explicitly model temporal consistency. Have you considered extensions that incorporate temporal constraints, such as recurrent mechanisms or temporal losses?
>
> **A2:** This is a fantastic question pointing to a valuable future direction. The reviewer is correct; our current work focuses on single-image processing, and **video results were generated frame-by-frame to show generalization.**
>
> We agree that explicit temporal modeling is the key to producing flicker-free videos. As part of our future research plan, we have considered several promising extensions for which our modular design is well-suited:
>
> 1. **Recurrent Mechanisms:** Integrating a ConvGRU to propagate temporal information between frames within our GTP module.
>
> 2. **Temporal Consistency Loss:** Applying a loss on the difference between consecutive output frames ($T_{t+1}, T_t, T_{t-1}$) to encourage smoothness.
>
> 3. **Optical Flow:** Using flow to warp previous frames or features, providing a stronger motion-aware consistency signal.
>
> We believe these extensions will further enhance DPRNet’s applicability to dynamic HDR video content, and we look forward to incorporating them in an extended version.
>
> ---
>
> > **Q3:** Could the proposed modules be generalized to other HDR-related tasks such as exposure fusion or inverse tone mapping?
>
> **A3:** Thank you for this excellent question about our method's broader significance. To demonstrate its generalizability, we have already evaluated DPRNet on the **inverse tone mapping (iTMO) task**. The results on three challenging benchmarks are shown below.
>
> **Table R2: Generalization to Inverse Tone Mapping**
>
> | Dataset             | Task                 | PSNR (dB) ↑ |   SSIM ↑  |
> |:--------------------|:---------------------|:---------:|:------:|
> | HDRPlus             | Inverse Tone Mapping |   31.94   | 0.9329 |
> | HDRI Haven          | Inverse Tone Mapping |   28.31   | 0.8953 |
> | HDRTVDM (2446c) | Inverse Tone Mapping |   33.40 | 0.9092 |
> | HDRTVDM (2446a) | Inverse Tone Mapping |  38.46 | 0.9536 |
>
> These strong results confirm that our core components are highly effective for iTMO. Specifically, our LDP is crucial for hallucinating lost high-frequency details, while the GTP/LTT framework excels at learning the complex, non-linear expansion curves. We believe this demonstrates the high potential of our modules for a range of HDR-related tasks.
>
> Once again, we are deeply thankful for your constructive feedback. Your suggestions have directly contributed to strengthening our work. Please do not hesitate to let us know if further clarification is needed.

---

> > ### Comment · Reviewer_nCUf · 2025-08-07
> >
> > The authors have provided a thoughtful and detailed response. The clarifications and additional analyses sufficiently address the concerns I raised. Including these clarifications in the final version would improve the overall clarity of the paper. I maintain my original score.

---

> > > ### Author Response · Authors · 2025-08-07
> > > **Acknowledgment to Reviewer nCUf**
> > >
> > > We are truly grateful for the reviewer's encouraging and positive assessment. We were very glad to learn that our rebuttal was considered "thoughtful and detailed" and that it successfully resolved your concerns. Thank you very much for your time and guidance. We fully commit to carefully revising the paper to include all the promised details, ensuring the final version is as clear and complete as possible. Thank you once again!

---

### Comment · Area_Chair_1RiX · 2025-08-01
**Author-Reviewer Discussion Period (July 31 - Aug 6)**

The author rebuttals are now posted.

To reviewers:
Please carefully read the *all* reviews and author responses, and engage in an open exchange with the authors.
Please post the response to the authors as soon as possible, so that we can have enough time for back-and-forth discussion with the authors.

---

> ### Comment · Area_Chair_1RiX · 2025-08-05
> **Discussion Period Ends Soon (Aug 6)!**
>
> Dear reviewers,
> Thanks so much for reviewing the paper. The discussion period ends soon. To ensure enough time to discuss this with the authors, please actively engage in the discussions with them if you have not done so.

---

### Note · Authors · 2025-08-12

**Dear Area Chair and Reviewers,**

We wish to express our most sincere gratitude to all reviewers for their invaluable time and deep insights throughout the review and discussion period.

We are delighted that, following a productive discussion, all reviewers (nCUf, wHV5, iFjE, aHBS) have confirmed that our responses have successfully addressed their concerns. Reviewer **nCUf** found our response "thoughtful and detailed" and confirmed that it "sufficiently address the concerns I raised." Likewise, Reviewer **wHV5** confirmed that "my concerns have been adequately addressed."

Furthermore, the in-depth engagement with Reviewers **iFjE** and **aHBS** provided us with the opportunity to clarify two core design principles of our work:

1. For the **GTP module**, we elaborated on its design rationale as a learnable generalization of classic global tone mapping operators.
2. For the **LTT module**, we detailed how its smooth, interpolation-based strategy fundamentally avoids artifacts from a "hard segmentation" approach.

As suggested by all reviewers, we formally commit to meticulously integrating all these clarifications and additional details into the final manuscript to significantly enhance its clarity. We are confident that these discussions have substantially strengthened our paper and have resolved all outstanding issues.

Thank you once again for your professional guidance; the paper is substantially stronger because of it.

---

### Decision · Program_Chairs · 2025-09-17

**Decision:**

Accept (poster)

**Comment:**

This paper proposes an end-to-end framework, Differential Pyramid Representation Network (DPRNet), for high-fidelity HDR image tone mapping. DPRNet incorporates a Learnable Differential Pyramid (LDP), along with Global Tone Perception (GTP) and Local Tone Tuning (LTT) modules, as well as Iterative Detail Enhancement (IDE). Experiments on four tone mapping datasets, including the new one proposed in the paper, show that the proposed method outperforms existing methods on standard HDR datasets in terms of PSNR and visual quality.

The authors valued the well-structured method for tone-mapping, as well as a new dataset called HDR Haven. The method is well-motivated in terms of tone mapping, such as well-known issues like halo artifacts and global-local tone imbalance.

The paper would be highly valuable for the community, considering both the methodological and strong dataset contributions.